# HOW ATTENTIVE ARE GRAPH ATTENTION NETWORKS?

**Shaked Brody**
Technion
shakedbr@cs.technion.ac.il

**Uri Alon**
Language Technologies Institute
Carnegie Mellon University
ualon@cs.cmu.edu

**Eran Yahav**
Technion
yahave@cs.technion.ac.il

## ABSTRACT

Graph Attention Networks (GATs) are one of the most popular GNN architectures and are considered as the state-of-the-art architecture for representation learning with graphs. In GAT, every node attends to its neighbors given its own representation as the query. However, in this paper we show that GAT computes a very limited kind of attention: the ranking of the attention scores is *unconditioned on the query node*. We formally define this restricted kind of attention as *static* attention and distinguish it from a strictly more expressive *dynamic* attention. Because GATs use a *static* attention mechanism, there are simple graph problems that GAT cannot express: in a controlled problem, we show that static attention hinders GAT from even fitting the training data. To remove this limitation, we introduce a simple fix by modifying the order of operations and propose GATv2: a *dynamic* graph attention variant that is strictly more expressive than GAT. We perform an extensive evaluation and show that GATv2 outperforms GAT across 12 OGB and other benchmarks while we match their parametric costs. Our code is available at https://github.com/tech-srl/how_attentive_are_gats.[1] GATv2 is available as part of the PyTorch Geometric library,[2] the Deep Graph Library,[3] and the TensorFlow GNN library.[4]

## 1 INTRODUCTION

Graph neural networks (GNNs; Gori et al., 2005; Scarselli et al., 2008) have seen increasing popularity over the past few years (Duvenaud et al., 2015; Atwood and Towsley, 2016; Bronstein et al., 2017; Monti et al., 2017). GNNs provide a general and efficient framework to learn from graph-structured data. Thus, GNNs are easily applicable in domains where the data can be represented as a set of nodes and the prediction depends on the relationships (edges) between the nodes. Such domains include molecules, social networks, product recommendation, computer programs and more.

In a GNN, each node iteratively updates its state by interacting with its neighbors. GNN variants (Wu et al., 2019; Xu et al., 2019; Li et al., 2016) mostly differ in how each node aggregates and combines the representations of its neighbors with its own. Veličković et al. (2018) pioneered the use of attention-based neighborhood aggregation, in one of the most common GNN variants – Graph Attention Network (GAT). In GAT, every node updates its representation by attending to its neighbors using its own representation as the query. This generalizes the standard averaging or max-pooling of neighbors (Kipf and Welling, 2017; Hamilton et al., 2017), by allowing every node to compute a *weighted* average of its neighbors, and (softly) select its most relevant neighbors. The work of

---

[1] An annotated implementation of GATv2 is available at https://nn.labml.ai/graphs/gatv2/
[2] from torch_geometric.nn.conv.gatv2_conv import GATv2Conv
[3] from dgl.nn.pytorch import GATv2Conv
[4] from tensorflow_gnn.graph.keras.layers.gat_v2 import GATv2Convolution

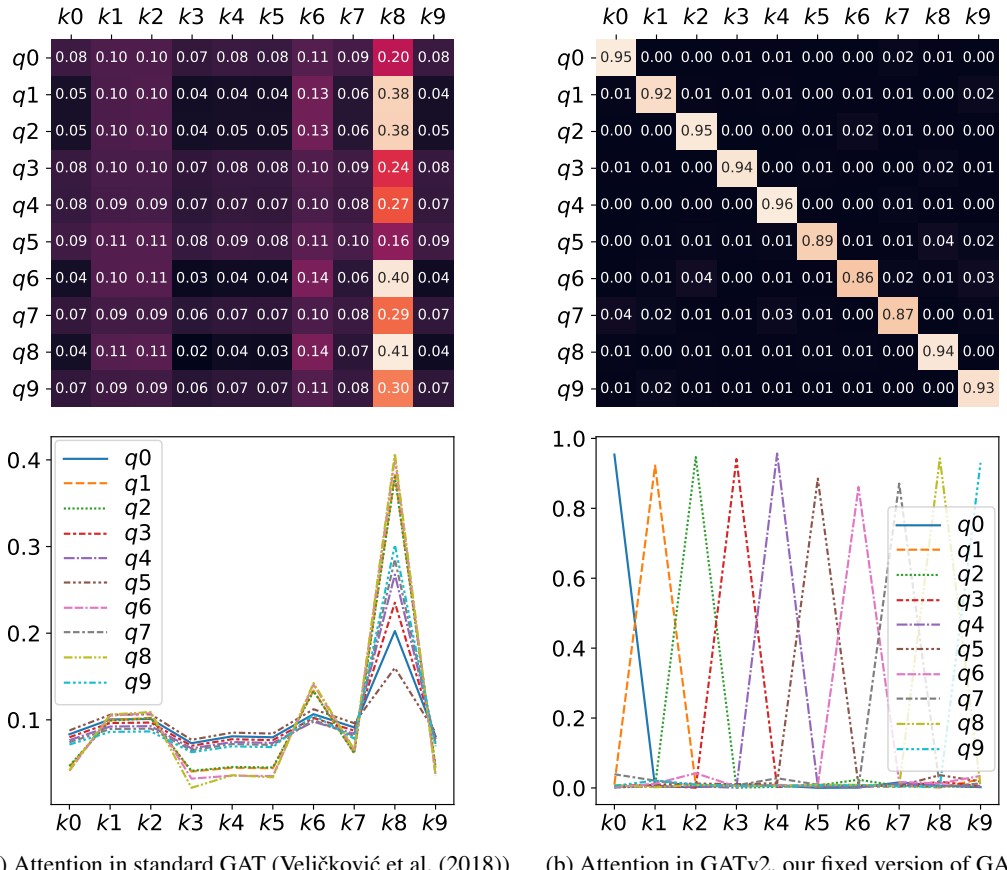

(a) Attention in standard GAT (Veličković et al. (2018))    (b) Attention in GATv2, our fixed version of GAT

Figure 1: In a complete bipartite graph of "query nodes" $\{q0, ..., q9\}$ and "key nodes" $\{k0, ..., k9\}$: standard GAT (Figure 1a) computes *static* attention – the ranking of attention coefficients is global for all nodes in the graph, and is unconditioned on the query node. For example, all queries ($q0$ to $q9$) attend mostly to the 8th key ($k8$). In contrast, GATv2 (Figure 1b) can actually compute *dynamic* attention, where every query has a different ranking of attention coefficients of the keys.

Veličković et al. also generalizes the Transformer's (Vaswani et al., 2017) self-attention mechanism, from sequences to graphs (Joshi, 2020).

Nowadays, GAT is one of the most popular GNN architectures (Bronstein et al., 2021) and is considered as the state-of-the-art neural architecture for learning with graphs (Wang et al., 2019a). Nevertheless, in this paper we show that *GAT does not actually compute the expressive, well known, type of attention* (Bahdanau et al., 2014), which we call *dynamic* attention. Instead, we show that GAT computes only a restricted "static" form of attention: for any query node, the attention function is *monotonic* with respect to the neighbor (key) scores. That is, the ranking (the $argsort$) of attention coefficients is identical for all nodes in the graph, and is *unconditioned* on the query node. This fact severely hurts the expressiveness of GAT, and is demonstrated in Figure 1a.

Supposedly, the conceptual idea of attention as the form of interaction between GNN nodes is orthogonal to the specific choice of attention function. However, Veličković et al.'s original design of GAT has spread to a variety of domains (Yang et al., 2020; Wang et al., 2019c; Huang and Carley, 2019; Ma et al., 2020; Kosaraju et al., 2019; Nathani et al., 2019; Wu et al., 2020; Zhang et al., 2020) and has become the default implementation of "graph attention network" in all popular GNN libraries such as PyTorch Geometric (Fey and Lenssen, 2019), DGL (Wang et al., 2019b), and others (Dwivedi et al., 2020; Gordić, 2020; Brockschmidt, 2020). Many other works employed GNNs with attention mechanisms other than the standard GAT's (Yun et al. (2019), see also Appendix A). However, none of these works identified the monotonicity of GAT's attention mechanism, the theoretical differences between attention types, nor empirically compared their performance.

To overcome the limitation we identified in GAT, we introduce a simple fix to its attention function by only modifying the order of internal operations. The result is GATv2 – a graph attention variant that has a universal approximator attention function, and is thus *strictly more expressive than GAT*. The effect of fixing the attention function in GATv2 is demonstrated in Figure 1b.

In summary, our main contribution is identifying that one of the most popular GNN types, the graph attention network, does not compute dynamic attention, the kind of attention that it seems to compute. We introduce formal definitions for analyzing the expressive power of graph attention mechanisms (Definitions 3.1 and 3.2), and derive our claims theoretically (Theorem 1) from the equations of Veličković et al. (2018). Empirically, we use a synthetic problem to show that standard GAT *cannot express* problems that require *dynamic* attention (Section 4.1). We introduce a simple fix by switching the order of internal operations in GAT, and propose GATv2, which *does* compute dynamic attention (Theorem 2). We further conduct a thorough empirical comparison of GAT and GATv2 and find that GATv2 outperforms GAT across 12 benchmarks of node-, link-, and graph-prediction. For example, GATv2 outperforms extensively tuned GNNs by over 1.4% in the difficult "UnseenProj Test" set of the VarMisuse task (Allamanis et al., 2018), without any hyperparameter tuning; and GATv2 improves over an extensively-tuned GAT by 11.5% in 13 prediction objectives in QM9. In node-prediction benchmarks from OGB (Hu et al., 2020), not only that GATv2 outperforms GAT with respect to accuracy – we find that dynamic attention provided a much better robustness to noise.

## 2 PRELIMINARIES

A directed graph $\mathcal{G} = (\mathcal{V}, \mathcal{E})$ contains nodes $\mathcal{V} = \{1, ..., n\}$ and edges $\mathcal{E} \subseteq \mathcal{V} \times \mathcal{V}$, where $(j, i) \in \mathcal{E}$ denotes an edge from a node $j$ to a node $i$. We assume that every node $i \in \mathcal{V}$ has an initial representation $\boldsymbol{h}_i^{(0)} \in \mathbb{R}^{d_0}$. An undirected graph can be represented with bidirectional edges.

### 2.1 GRAPH NEURAL NETWORKS

A graph neural network (GNN) layer updates every node representation by aggregating its neighbors' representations. A layer's input is a set of node representations $\{\boldsymbol{h}_i \in \mathbb{R}^d \mid i \in \mathcal{V}\}$ and the set of edges $\mathcal{E}$. A layer outputs a new set of node representations $\{\boldsymbol{h}_i' \in \mathbb{R}^{d'} \mid i \in \mathcal{V}\}$, where the same parametric function is applied to every node given its neighbors $\mathcal{N}_i = \{j \in \mathcal{V} \mid (j, i) \in \mathcal{E}\}$:

$$\boldsymbol{h}_i' = f_\theta \left( \boldsymbol{h}_i, \mathrm{AGGREGATE} \left( \{\boldsymbol{h}_j \mid j \in \mathcal{N}_i\} \right) \right) \tag{1}$$

The design of $f$ and AGGREGATE is what mostly distinguishes one type of GNN from the other. For example, a common variant of GraphSAGE (Hamilton et al., 2017) performs an element-wise mean as AGGREGATE, followed by concatenation with $\boldsymbol{h}_i$, a linear layer and a ReLU as $f$.

### 2.2 GRAPH ATTENTION NETWORKS

GraphSAGE and many other popular GNN architectures (Xu et al., 2019; Duvenaud et al., 2015) weigh all neighbors $j \in \mathcal{N}_i$ with *equal importance* (e.g., mean or max-pooling as AGGREGATE). To address this limitation, GAT (Veličković et al., 2018) instantiates Equation (1) by computing a learned weighted average of the representations of $\mathcal{N}_i$. A scoring function $e : \mathbb{R}^d \times \mathbb{R}^d \to \mathbb{R}$ computes a score for every edge $(j, i)$, which indicates the importance of the features of the neighbor $j$ to the node $i$:

$$e \left( \boldsymbol{h}_i, \boldsymbol{h}_j \right) = \mathrm{LeakyReLU} \left( \boldsymbol{a}^\top \cdot [\boldsymbol{W}\boldsymbol{h}_i \| \boldsymbol{W}\boldsymbol{h}_j] \right) \tag{2}$$

where $\boldsymbol{a} \in \mathbb{R}^{2d'}$, $\boldsymbol{W} \in \mathbb{R}^{d' \times d}$ are learned, and $\|$ denotes vector concatenation. These attention scores are normalized across all neighbors $j \in \mathcal{N}_i$ using softmax, and the attention function is defined as:

$$\alpha_{ij} = \mathrm{softmax}_j \left( e \left( \boldsymbol{h}_i, \boldsymbol{h}_j \right) \right) = \frac{\exp \left( e \left( \boldsymbol{h}_i, \boldsymbol{h}_j \right) \right)}{\sum_{j' \in \mathcal{N}_i} \exp \left( e \left( \boldsymbol{h}_i, \boldsymbol{h}_{j'} \right) \right)} \tag{3}$$

Then, GAT computes a weighted average of the transformed features of the neighbor nodes (followed by a nonlinearity $\sigma$) as the new representation of $i$, using the normalized attention coefficients:

$$\boldsymbol{h}_i' = \sigma \left( \sum_{j \in \mathcal{N}_i} \alpha_{ij} \cdot \boldsymbol{W}\boldsymbol{h}_j \right) \tag{4}$$

From now on, we will refer to Equations (2) to (4) as the definition of GAT.

## 3 THE EXPRESSIVE POWER OF GRAPH ATTENTION MECHANISMS

In this section, we explain why attention is limited when it is not *dynamic* (Section 3.1). We then show that GAT is severely constrained, because it can only compute *static* attention (Section 3.2). Next, we show how GAT can be fixed (Section 3.3), by simply modifying the order of operations.

We refer to a neural architecture (e.g., the scoring or the attention function of GAT) as a *family of functions*, parameterized by the learned parameters. An element in the family is a concrete function with specific trained weights. In the following, we use $[n]$ to denote the set $[n] = \{1, 2, ..., n\} \subset \mathbb{N}$.

### 3.1 THE IMPORTANCE OF DYNAMIC WEIGHTING

Attention is a mechanism for computing a distribution over a set of input *key* vectors, given an additional *query* vector. If the attention function always weighs one key at least as much as any other key, *unconditioned on the query*, we say that this attention function is *static*:

**Definition 3.1** (Static attention). A (possibly infinite) family of scoring functions $\mathcal{F} \subseteq \left( \mathbb{R}^d \times \mathbb{R}^d \to \mathbb{R} \right)$ computes *static scoring* for a given set of key vectors $\mathbb{K} = \{\boldsymbol{k}_1, ..., \boldsymbol{k}_n\} \subset \mathbb{R}^d$ and query vectors $\mathbb{Q} = \{\boldsymbol{q}_1, ..., \boldsymbol{q}_m\} \subset \mathbb{R}^d$, if for every $f \in \mathcal{F}$ there exists a "highest scoring" key $j_f \in [n]$ such that for every query $i \in [m]$ and key $j \in [n]$ it holds that $f\left(\boldsymbol{q}_i, \boldsymbol{k}_{j_f}\right) \geq f\left(\boldsymbol{q}_i, \boldsymbol{k}_j\right)$. We say that a family of attention functions computes *static attention* given $\mathbb{K}$ and $\mathbb{Q}$, if its scoring function computes static scoring, possibly followed by monotonic normalization such as softmax.

Static attention is very limited because every function $f \in \mathcal{F}$ has a key that is *always selected*, regardless of the query. Such functions cannot model situations where different keys have different relevance to different queries. Static attention is demonstrated in Figure 1a.

The general and powerful form of attention is *dynamic attention*:

**Definition 3.2** (Dynamic attention). A (possibly infinite) family of scoring functions $\mathcal{F} \subseteq \left( \mathbb{R}^d \times \mathbb{R}^d \to \mathbb{R} \right)$ computes *dynamic scoring* for a given set of key vectors $\mathbb{K} = \{\boldsymbol{k}_1, ..., \boldsymbol{k}_n\} \subset \mathbb{R}^d$ and query vectors $\mathbb{Q} = \{\boldsymbol{q}_1, ..., \boldsymbol{q}_m\} \subset \mathbb{R}^d$, if for *any* mapping $\varphi \colon [m] \to [n]$ there exists $f \in \mathcal{F}$ such that for any query $i \in [m]$ and any key $j_{\neq \varphi(i)} \in [n]$: $f\left(\boldsymbol{q}_i, \boldsymbol{k}_{\varphi(i)}\right) > f\left(\boldsymbol{q}_i, \boldsymbol{k}_j\right)$. We say that a family of attention functions computes *dynamic attention* for $\mathbb{K}$ and $\mathbb{Q}$, if its scoring function computes dynamic scoring, possibly followed by monotonic normalization such as softmax.

That is, dynamic attention can *select* every key $\varphi(i)$ using the query $i$, by making $f\left(\boldsymbol{q}_i, \boldsymbol{k}_{\varphi(i)}\right)$ the maximal in $\{f\left(\boldsymbol{q}_i, \boldsymbol{k}_j\right) \mid j \in [n]\}$. Note that *dynamic* and *static* attention are exclusive properties, but they are not complementary. Further, every *dynamic* attention family has strict subsets of *static* attention families with respect to the same $\mathbb{K}$ and $\mathbb{Q}$. Dynamic attention is demonstrated in Figure 1b.

**Attending by decaying** Another way to think about attention is the ability to "focus" on the most relevant inputs, given a query. Focusing is only possible by *decaying* other inputs, i.e., giving these decayed inputs lower scores than others. If one key is always given an equal or greater attention score than other keys (as in static attention), no query can ignore this key or decay this key's score.

### 3.2 THE LIMITED EXPRESSIVITY OF GAT

Although the scoring function $e$ can be defined in various ways, the original definition of Veličković et al. (2018) (Equation (2)) has become the *de facto* practice: it has spread to a variety of domains and is now the standard implementation of "graph attention network" in all popular GNN libraries (Fey and Lenssen, 2019; Wang et al., 2019b; Dwivedi et al., 2020; Gordić, 2020; Brockschmidt, 2020).

The motivation of GAT is to compute a representation for every node as a weighted average of its neighbors. Statedly, GAT is inspired by the attention mechanism of Bahdanau et al. (2014) and the self-attention mechanism of the Transformer (Vaswani et al., 2017). Nonetheless:

**Theorem 1.** *A GAT layer computes only static attention, for any set of node representations* $\mathbb{K} = \mathbb{Q} = \{\boldsymbol{h}_1, ..., \boldsymbol{h}_n\}$. *In particular, for $n > 1$, a GAT layer does not compute dynamic attention.*

*Proof.* Let $\mathcal{G} = (\mathcal{V}, \mathcal{E})$ be a graph modeled by a GAT layer with some $\boldsymbol{a}$ and $\boldsymbol{W}$ values (Equations (2) and (3)), and having node representations $\{\boldsymbol{h}_1, ..., \boldsymbol{h}_n\}$. The learned parameter $\boldsymbol{a}$ can be written as a

concatenation $\boldsymbol{a} = [\boldsymbol{a}_1 \| \boldsymbol{a}_2] \in \mathbb{R}^{2d'}$ such that $\boldsymbol{a}_1, \boldsymbol{a}_2 \in \mathbb{R}^{d'}$, and Equation (2) can be re-written as:

$$e(\boldsymbol{h}_i, \boldsymbol{h}_j) = \text{LeakyReLU}\left(\boldsymbol{a}_1^\top \boldsymbol{W} \boldsymbol{h}_i + \boldsymbol{a}_2^\top \boldsymbol{W} \boldsymbol{h}_j\right) \tag{5}$$

Since $\mathcal{V}$ is finite, there exists a node $j_{max} \in \mathcal{V}$ such that $\boldsymbol{a}_2^\top \boldsymbol{W} \boldsymbol{h}_{j_{max}}$ is maximal among all nodes $j \in \mathcal{V}$ ($j_{max}$ is the $j_f$ required by Definition 3.1). Due to the monotonicity of LeakyReLU and softmax, for every query node $i \in \mathcal{V}$, the node $j_{max}$ also leads to the maximal value of its attention distribution $\{\alpha_{ij} \mid j \in \mathcal{V}\}$. Thus, from Definition 3.1 directly, $\alpha$ computes only *static attention*. This also implies that $\alpha$ does not compute dynamic attention, because in GAT, Definition 3.2 holds only for *constant* mappings $\varphi$ that map all inputs to the same output. $\qquad\square$

The consequence of Theorem 1 is that for any set of nodes $\mathcal{V}$ and a trained GAT layer, the attention function $\alpha$ defines a constant ranking ($argsort$) of the nodes, unconditioned on the query nodes $i$. That is, we can denote $s_j = \boldsymbol{a}_2^\top \boldsymbol{W} \boldsymbol{h}_j$ and get that for any choice of $\boldsymbol{h}_i$, $\alpha$ is monotonic with respect to the per-node scores $\{s_j \mid j \in \mathcal{V}\}$. This global ranking induces the local ranking of every neighborhood $\mathcal{N}_i$. The only effect of $\boldsymbol{h}_i$ is in the "sharpness" of the produced attention distribution. This is demonstrated in Figure 1a (bottom), where different curves denote different queries ($\boldsymbol{h}_i$). A discussion regarding the generalization to multi-head attention can be found in Appendix C.

### 3.3 Building Dynamic Graph Attention Networks

To create a *dynamic* graph attention network, we modify the order of internal operations in GAT and introduce GATv2 – a simple fix of GAT that has a strictly more expressive attention mechanism.

**GATv2** The main problem in the standard GAT scoring function (Equation (2)) is that the learned layers $\boldsymbol{W}$ and $\boldsymbol{a}$ are applied consecutively, and thus can be collapsed into a *single* linear layer. To fix this limitation, we simply apply the $\boldsymbol{a}$ layer *after* the nonlinearity (LeakyReLU), and the $\boldsymbol{W}$ layer after the concatenation,[5] effectively applying an MLP to compute the score for each query-key pair:

$$\text{GAT (Veličković et al., 2018):} \qquad e(\boldsymbol{h}_i, \boldsymbol{h}_j) = \text{LeakyReLU}\left(\boldsymbol{a}^\top \cdot [\boldsymbol{W}\boldsymbol{h}_i \| \boldsymbol{W}\boldsymbol{h}_j]\right) \tag{6}$$

$$\text{GATv2 (our fixed version):} \qquad e(\boldsymbol{h}_i, \boldsymbol{h}_j) = \boldsymbol{a}^\top \text{LeakyReLU}\left(\boldsymbol{W} \cdot [\boldsymbol{h}_i \| \boldsymbol{h}_j]\right) \tag{7}$$

The simple modification makes a significant difference in the expressiveness of the attention function:

**Theorem 2.** *A GATv2 layer computes dynamic attention for any set of node representations* $\mathbb{K} = \mathbb{Q} = \{\boldsymbol{h}_1, ..., \boldsymbol{h}_n\}$.

We prove Theorem 2 in Appendix B. The main idea is that we can define an appropriate function that GATv2 will be a universal approximator (Cybenko, 1989; Hornik, 1991) of. In contrast, GAT (Equation (52)) cannot approximate any such desired function (Theorem 1).

**Complexity** GATv2 has the same time-complexity as GAT's declared complexity: $\mathcal{O}\left(|\mathcal{V}|dd' + |\mathcal{E}|d'\right)$. However, by merging its linear layers, GAT can be computed faster than stated by Veličković et al. (2018). For a detailed time- and parametric-complexity analysis, see Appendix I.

## 4 Evaluation

First, we demonstrate the weakness of GAT using a simple synthetic problem that GAT cannot even fit (cannot even achieve high *training* accuracy), but is easily solvable by GATv2 (Section 4.1). Second, we show that GATv2 is much more *robust to edge noise*, because its dynamic attention mechanisms allow it to decay noisy (false) edges, while GAT's performance severely decreases as noise increases (Section 4.2). Finally, we compare GAT and GATv2 across 12 benchmarks overall. (Sections 4.3 to 4.6 and appendix F.4). We find that GAT is inferior to GATv2 across all examined benchmarks.

**Setup** When previous results exist, we take hyperparameters that were tuned for GAT and use them in GATv2, without any additional tuning. Self-supervision (Kim and Oh, 2021; Rong et al., 2020a), graph regularization (Zhao and Akoglu, 2020; Rong et al., 2020b), and other tricks (Wang, 2021; Huang et al., 2021) are orthogonal to the contribution of the GNN layer itself, and may further improve

---

[5]We also add a bias vector $\boldsymbol{b}$ before applying the nonlinearity, we omit this in Equation (7) for brevity.

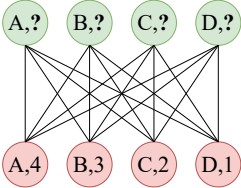

Figure 2: The DICTIONARY-LOOKUP problem of size $k$=4: every node in the bottom row has an alphabetic *attribute* ($\{A, B, C, ...\}$) and a numeric *value* ($\{1, 2, 3, ...\}$); every node in the upper row has only an attribute; the goal is to predict the value for each node in the upper row, using its attribute.

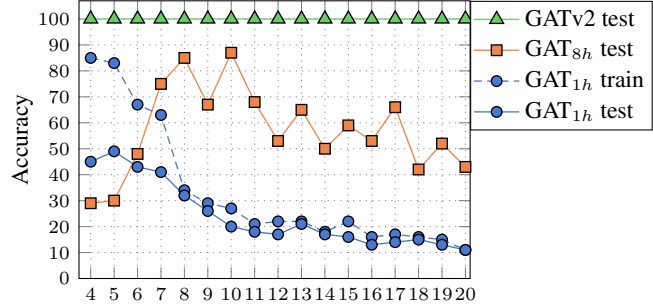

$k$ (number of different keys in each graph)

Figure 3: The DICTIONARYLOOKUP problem: GATv2 easily achieves 100% train and test accuracies even for $k$=100 and using only a single head.

all GNNs. In all experiments of GATv2, we constrain the learned matrix by setting $\boldsymbol{W} = [\boldsymbol{W}' \| \boldsymbol{W}']$, to rule out the increased number of parameters over GAT as the source of empirical difference (see Appendix I.2). Training details, statistics, and code are provided in Appendix D.

Our main goal is to compare dynamic and static graph attention mechanisms. However, for reference, we also include non-attentive baselines such as GCN (Kipf and Welling, 2017), GIN (Xu et al., 2019) and GraphSAGE (Hamilton et al., 2017). These non-attentive GNNs can be thought of as a special case of attention, where every node gives all its neighbors the same attention score. Additional comparison to a Transformer-style scaled dot-product attention ("DPGAT"), which is *strictly weaker* than our proposed GATv2 (see a proof in Appendix G.1), is shown in Appendix G.

## 4.1 SYNTHETIC BENCHMARK: DICTIONARYLOOKUP

The DICTIONARYLOOKUP problem is a contrived problem that we designed to test the ability of a GNN architecture to perform dynamic attention. Here, we demonstrate that GAT cannot learn this simple problem. Figure 2 shows a complete bipartite graph of $2k$ nodes. Each "key node" in the bottom row has an *attribute* ($\{A, B, C, ...\}$) and a *value* ($\{1, 2, 3, ...\}$). Each "query node" in the upper row has *only an attribute* ($\{A, B, C, ...\}$). The goal is to predict the value of every query node (upper row), according to its attribute. Each graph in the dataset has a different mapping from attributes to values. We created a separate dataset for each $k = \{1, 2, 3, ...\}$, for which we trained a different model, and measured per-node accuracy.

Although this is a contrived problem, it is relevant to any subgraph with keys that share more than one query, and each query needs to attend to the keys differently. Such subgraphs are very common in a variety of real-world domains. This problem tests the layer itself because it can be solved using a *single* GNN layer, without suffering from multi-layer side-effects such as over-smoothing (Li et al., 2018), over-squashing (Alon and Yahav, 2021), or vanishing gradients (Li et al., 2019). Our code will be made publicly available, to serve as a testbed for future graph attention mechanisms.

**Results** Figure 3 shows the following surprising results: GAT with a single head ($GAT_{1h}$) failed to fit the *training* set for any value of $k$, no matter for how many iterations it was trained, and after trying various training methods. Thus, it expectedly fails to generalize (resulting in low test accuracy). Using 8 heads, $GAT_{8h}$ successfully fits the *training* set, but generalizes *poorly* to the *test* set. In contrast, GATv2 easily achieves 100% training and 100% test accuracies for any value of $k$, and even for $k$=100 (not shown) and using a *single head*, thanks to its ability to perform dynamic attention. These results clearly show the limitations of GAT, which are easily solved by GATv2. An additional comparison to GIN, which could *not* fit this dataset, is provided in Figure 5 in Appendix F.1.

**Visualization** Figure 1a (top) shows a heatmap of GAT's attention scores in this DICTIONARY-LOOKUP problem. As shown, all query nodes $q0$ to $q9$ attend mostly to the eighth key ($k8$), and have the same ranking of attention coefficients (Figure 1a (bottom)). In contrast, Figure 1b shows how GATv2 can *select* a different key node for every query node, because it computes dynamic attention.

**The role of multi-head attention** Veličković et al. (2018) found the role of multi-head attention to be stabilizing the learning process. Nevertheless, Figure 3 shows that increasing the number of heads strictly increases training accuracy, and thus, the expressivity. Thus, GAT *depends* on having multiple attention heads. In contrast, even a *single* GATv2 head generalizes better than a multi-head GAT.

## 4.2 ROBUSTNESS TO NOISE

We examine the robustness of *dynamic* and *static* attention to noise. In particular, we focus on structural noise: given an input graph $\mathcal{G} = (\mathcal{V}, \mathcal{E})$ and a noise ratio $0 \leq p \leq 1$, we randomly sample $|\mathcal{E}| \times p$ non-existing edges $\mathcal{E}'$ from $\mathcal{V} \times \mathcal{V} \setminus \mathcal{E}$. We then train the GNN on the noisy graph $\mathcal{G}' = (\mathcal{V}, \mathcal{E} \cup \mathcal{E}')$.

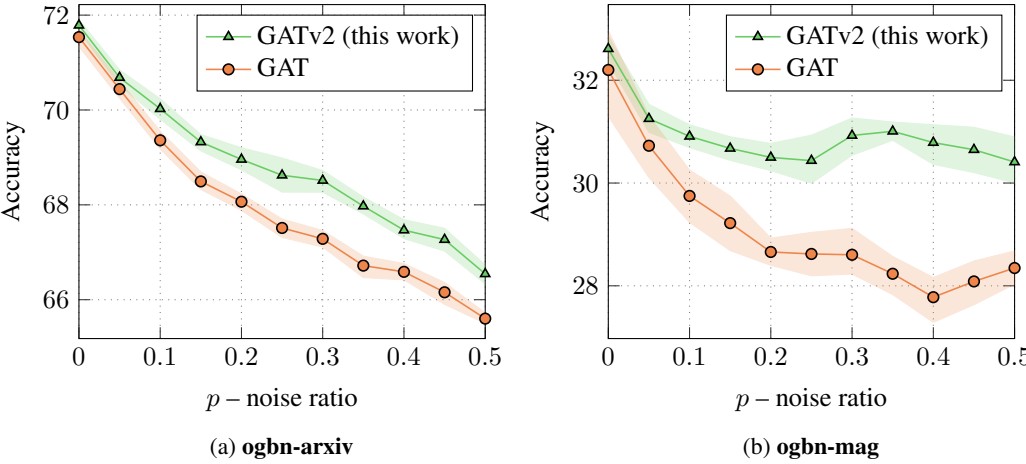

(a) **ogbn-arxiv**    (b) **ogbn-mag**

Figure 4: Test accuracy compared to the noise ratio: GATv2 is more robust to structural noise compared to GAT. Each point is an average of 10 runs, error bars show standard deviation.

**Results** Figure 8 shows the accuracy on two node-prediction datasets from the Open Graph Benchmark (OGB; Hu et al., 2020) as a function of the noise ratio $p$. As $p$ increases, all models show a natural decline in test accuracy in both datasets. Yet, thanks to their ability to compute *dynamic* attention, GATv2 shows a milder degradation in accuracy compared to GAT, which shows a steeper descent. We hypothesize that the ability to perform *dynamic* attention helps the models distinguishing between given data edges ($\mathcal{E}$) and noise edges ($\mathcal{E}'$); in contrast, GAT cannot distinguish between edges, because it scores the source and target nodes separately. These results clearly demonstrate the *robustness* of *dynamic* attention over *static* attention in noisy settings, which are common in reality.

## 4.3 PROGRAMS: VARMISUSE

**Setup** VARMISUSE (Allamanis et al., 2018) is an inductive node-pointing problem that depends on 11 types of syntactic and semantic interactions between elements in computer programs.

We used the framework of Brockschmidt (2020), who performed an extensive hyperparameter tuning by searching over 30 configurations for every GNN type. We took their best GAT hyperparameters and used them to train GATv2, without further tuning.

**Results** As shown in Table 1, GATv2 is more accurate than GAT and other GNNs in the SeenProj test sets. Furthermore, GATv2 achieves an even higher improvement in the *Unseen*Proj test set. Overall, these results demonstrate the power of GATv2 in modeling complex relational problems, especially

Table 1: Accuracy (5 runs±stdev) on VARMISUSE. GATv2 is more accurate than all GNNs in both test sets, using GAT's hyperparameters. † previously reported by Brockschmidt (2020).

|  | Model | SeenProj | UnseenProj |
|---|---|---|---|
| No-Attention | GCN[†] | 87.2±1.5 | 81.4±2.3 |
|  | GIN[†] | 87.1±0.1 | 81.1±0.9 |
| Attention | GAT[†] | 86.9±0.7 | 81.2±0.9 |
|  | GATv2 | **88.0**±1.1 | **82.8**±1.7 |

since it outperforms extensively tuned models, without any further tuning by us.

## 4.4 Node-Prediction

We further compare GATv2, GAT, and other GNNs on four node-prediction datasets from OGB.

Table 2: Average accuracy (Table 2a) and ROC-AUC (Table 2b) in node-prediction datasets (10 runs±std). In all datasets, GATv2 outperforms GAT. † – previously reported by Hu et al. (2020).

| | | (a) | | | (b) |
|---|---|---|---|---|---|
| Model | Attn. Heads | ogbn-arxiv | ogbn-products | ogbn-mag | ogbn-proteins |
| GCN[†] | 0 | $71.74_{\pm0.29}$ | $78.97_{\pm0.33}$ | $30.43_{\pm0.25}$ | $72.51_{\pm0.35}$ |
| GraphSAGE[†] | 0 | $71.49_{\pm0.27}$ | $78.70_{\pm0.36}$ | $31.53_{\pm0.15}$ | $77.68_{\pm0.20}$ |
| GAT | 1 | $71.59_{\pm0.38}$ | $79.04_{\pm1.54}$ | $32.20_{\pm1.46}$ | $70.77_{\pm5.79}$ |
| | 8 | $71.54_{\pm0.30}$ | $77.23_{\pm2.37}$ | $31.75_{\pm1.60}$ | $78.63_{\pm1.62}$ |
| GATv2 (this work) | 1 | $71.78_{\pm0.18}$ | $\mathbf{80.63}_{\pm0.70}$ | $\mathbf{32.61}_{\pm0.44}$ | $77.23_{\pm3.32}$ |
| | 8 | $\mathbf{71.87}_{\pm0.25}$ | $78.46_{\pm2.45}$ | $32.52_{\pm0.39}$ | $\mathbf{79.52}_{\pm0.55}$ |

**Results** Results are shown in Table 2. In all settings and all datasets, GATv2 is more accurate than GAT and the non-attentive GNNs. Interestingly, in the datasets of Table 2a, *even a single head of GATv2 outperforms GAT with 8 heads*. In Table 2b (**ogbn-proteins**), increasing the number of heads results in a major improvement for GAT (from 70.77 to 78.63), while GATv2 already gets most of the benefit using a single attention head. These results demonstrate the superiority of GATv2 over GAT in node prediction (and even with a single head), thanks to GATv2's dynamic attention.

## 4.5 Graph-Prediction: QM9

**Setup** In the QM9 dataset (Ramakrishnan et al., 2014; Gilmer et al., 2017), each graph is a molecule and the goal is to regress each graph to 13 real-valued quantum chemical properties. We used the implementation of Brockschmidt (2020) who performed an extensive hyperparameter search over 500 configurations; we took their best-found configuration of GAT to implement GATv2.

Table 3: Average error rates (lower is better), 5 runs for each property, on the QM9 dataset. The best result among GAT and GATv2 is marked in **bold**; the globally best result among all GNNs is marked in **bold and underline**. † was previously tuned and reported by Brockschmidt (2020).

| | Predicted Property | | | | | | | | | | | | | Rel. to |
|---|---|---|---|---|---|---|---|---|---|---|---|---|---|---|
| Model | 1 | 2 | 3 | 4 | 5 | 6 | 7 | 8 | 9 | 10 | 11 | 12 | 13 | GAT |
| GCN[†] | 3.21 | **4.22** | 1.45 | 1.62 | 2.42 | 16.38 | 17.40 | 7.82 | 8.24 | 9.05 | 7.00 | 3.93 | **1.02** | -1.5% |
| GIN[†] | **2.64** | 4.67 | 1.42 | 1.50 | **2.27** | **15.63** | **12.93** | **5.88** | 18.71 | **5.62** | **5.38** | **3.53** | 1.05 | -2.3% |
| GAT[†] | 2.68 | 4.65 | 1.48 | 1.53 | 2.31 | 52.39 | 14.87 | 7.61 | 6.86 | 7.64 | 6.54 | 4.11 | **1.48** | +0% |
| GATv2 | **2.65** | **4.28** | **1.41** | **1.47** | **2.29** | **16.37** | **14.03** | **6.07** | **6.28** | **6.60** | **5.97** | **3.57** | 1.59 | **-11.5%** |

**Results** Table 3 shows the main results: GATv2 achieves a lower (better) average error than GAT, by 11.5% relatively. GAT achieves the overall highest average error. In some properties, the non-attentive GNNs, GCN and GIN, perform best. We hypothesize that attention is not needed in modeling these properties. Generally, GATv2 achieves the lowest overall average relative error (rightmost column).

## 4.6 Link-Prediction

We compare GATv2, GAT, and other GNNs in link-prediction datasets from OGB.

**Results** Table 8 shows that in all datasets, GATv2 achieves a higher MRR than GAT, which achieves the lowest MRR. However, the non-attentive GraphSAGE performs better than all attentive GNNs. We hypothesize that attention might not be needed in these datasets. Another possibility is that dynamic attention is especially useful in graphs that have *high node degrees*: in **ogbn-products** and **ogbn-proteins** (Table 2) the average node degrees are 50.5 and 597, respectively (see Table 5 in

Appendix E). **ogbl-collab** and **ogbl-citation2** (Table 8), however, have much lower average node degrees – of 8.2 and 20.7. We hypothesize that a dynamic attention mechanism is especially useful to select the most relevant neighbors when the total number of neighbors is high. We leave the study of the effect of the datasets's average node degrees on the optimal GNN architecture for future work.

### 4.7 DISCUSSION

In *all* examined benchmarks, we found that *GATv2 is more accurate than GAT*. Further, we found that GATv2 is significantly more robust to noise than GAT. In the synthetic DICTIONARYLOOKUP benchmark (Section 4.1), GAT fails to express the data, and thus achieves even poor *training* accuracy.

In few of the benchmarks (Table 8 and some of the properties in Table 3) – a non-attentive model such as GCN or GIN achieved a higher accuracy than all GNNs that do use attention. We hypothesize that attention is not needed in these datasets.

**Which graph attention mechanism should I use?** It is usually impossible to determine in advance which architecture would perform best. A theoretically weaker model may perform better in practice, because a stronger model might overfit the training data if the task is "too simple" and does not require such expressiveness. Intuitively, we believe that the more complex the interactions between nodes are – the more benefit a GNN can take from theoretically stronger graph attention mechanisms such as GATv2. The main question is whether the problem has a *global ranking* of "influential" nodes (GAT is sufficient), or do different nodes have *different rankings* of neighbors (use GATv2).

Veličković, the author of GAT, has confirmed on Twitter[6] that GAT was designed to work in the "easy-to-overfit" datasets of the time (2017), such as Cora, Citeseer and Pubmed (Sen et al., 2008), where the data might had an underlying static ranking of "globally important" nodes. Veličković agreed that newer and more challenging benchmarks may demand stronger attention mechanisms such as GATv2. that depends on the representation of the query node as well. In this paper, we revisit the traditional assumptions and show that many modern graph benchmarks and datasets contain more complex interactions, and thus *require dynamic attention*.

## 5 CONCLUSION

In this paper, we identify that the popular and widespread Graph Attention Network does not compute *dynamic* attention. Instead, the attention mechanism in the standard definition and implementations of GAT is only *static*: for any query, its neighbor-scoring is monotonic with respect to per-node scores. Further, the ranking (the $argsort$) of attention coefficients is identical for all nodes in the graph, and is unconditioned on the query node. As a result, GAT cannot even express simple alignment problems. To address this limitation, we introduce a simple fix and propose GATv2: by modifying the order of operations in GAT, GATv2 achieves a universal approximator attention function and is thus strictly more powerful than GAT.

We demonstrate the empirical advantage of GATv2 over GAT in a synthetic problem that requires dynamic selection of nodes, and in 12 benchmarks from OGB and other public datasets. Our experiments show that GATv2 outperforms GAT in all benchmarks while having the same parametric cost.

We encourage the community to use GATv2 instead or in addition to GAT whenever comparing new GNN architectures to the common strong baselines. In complex tasks and domains and in challenging datasets, a model that uses GAT as an internal component can replace it with GATv2 to benefit from a strictly more powerful model. To this end, we make our code publicly available at `https://github.com/tech-srl/how_attentive_are_gats`,[7] and GATv2 is available as part of the PyTorch Geometric library,[8] the Deep Graph Library,[9] and the TensorFlow GNN library.[10]

---

[6]`https://twitter.com/PetarV_93/status/1399685979506675714`

[7]An annotated implementation of GATv2 is available at `https://nn.labml.ai/graphs/gatv2/`

[8]`from torch_geometric.nn.conv.gatv2_conv import GATv2Conv`

[9]`from dgl.nn.pytorch import GATv2Conv`

[10]`from tensorflow_gnn.graph.keras.layers.gat_v2 import GATv2Convolution`

## ACKNOWLEDGMENTS

We thank Gail Weiss for the helpful discussions, thorough feedback, and inspirational paper (Weiss et al., 2018). We also thank Petar Veličković for the useful initial discussion about the complexity and implementation of GAT, Lucio Dery, and the Graph Representation Learning Reading Group at Mila[11] for their useful comments and questions.

## A  RELATED WORK

**Attention in GNNs** Modeling pairwise interactions between elements in graph-structured data goes back to interaction networks (Battaglia et al., 2016; Hoshen, 2017) and relational networks (Santoro et al., 2017). The GAT formulation of Veličković et al. (2018) rose as the most popular framework for attentional GNNs, thanks to its simplicity, generality, and applicability beyond reinforcement learning (Denil et al., 2017; Duan et al., 2017). Nevertheless, in this work, we show that the popular and widespread definition of GAT is severely constrained to static attention only.

**Other graph attention mechanisms** Many works employed GNNs with attention mechanisms other than the standard GAT's (Zhang et al., 2018; Thekumparampil et al., 2018; Gao and Ji, 2019; Lukovnikov and Fischer, 2021; Shi et al., 2020; Dwivedi and Bresson, 2020; Busbridge et al., 2019; Rong et al., 2020a; Veličković et al., 2020), and Lee et al. (2018) conducted an extensive survey of attention types in GNNs. However, none of these works identified the monotonicity of GAT's attention mechanism, the theoretical differences between attention types, nor empirically compared their performance. Kim and Oh (2021) compared two graph attention mechanisms empirically, but in a specific self-supervised scenario, without observing the theoretical difference in their expressiveness.

**The static attention of GAT** Qiu et al. (2018) recognized the order-preserving property of GAT, but did not identify the severe theoretical constraint that this property implies: the inability to perform dynamic attention (Theorem 1). Furthermore, they presented GAT's monotonicity as a *desired* trait (!) To the best of our knowledge, our work is the first work to recognize the inability of GAT to perform dynamic attention and its practical harmful consequences.

## B  PROOF FOR THEOREM 2

For brevity, we repeat our definition of dynamic attention (Definition 3.2):

**Definition 3.2** (Dynamic attention). A (possibly infinite) family of scoring functions $\mathcal{F} \subseteq \left(\mathbb{R}^d \times \mathbb{R}^d \to \mathbb{R}\right)$ computes *dynamic scoring* for a given set of key vectors $\mathbb{K} = \{\boldsymbol{k}_1, ..., \boldsymbol{k}_n\} \subset \mathbb{R}^d$ and query vectors $\mathbb{Q} = \{\boldsymbol{q}_1, ..., \boldsymbol{q}_m\} \subset \mathbb{R}^d$, if for *any* mapping $\varphi \colon [m] \to [n]$ there exists $f \in \mathcal{F}$ such that for any query $i \in [m]$ and any key $j_{\neq \varphi(i)} \in [n]$: $f\left(\boldsymbol{q}_i, \boldsymbol{k}_{\varphi(i)}\right) > f\left(\boldsymbol{q}_i, \boldsymbol{k}_j\right)$. We say that a family of attention functions computes *dynamic attention* for $\mathbb{K}$ and $\mathbb{Q}$, if its scoring function computes dynamic scoring, possibly followed by monotonic normalization such as softmax.

**Theorem 2.** *A GATv2 layer computes dynamic attention for any set of node representations $\mathbb{K} = \mathbb{Q} = \{\boldsymbol{h}_1, ..., \boldsymbol{h}_n\}$.*

*Proof.* Let $\mathcal{G} = (\mathcal{V}, \mathcal{E})$ be a graph modeled by a GATv2 layer, having node representations $\{\boldsymbol{h}_1, ..., \boldsymbol{h}_n\}$, and let $\varphi \colon [n] \to [n]$ be any node mapping $[n] \to [n]$. We define $g \colon \mathbb{R}^{2d} \to \mathbb{R}$ as follows:

$$g\left(\boldsymbol{x}\right) = \begin{cases} 1 & \exists i \colon \boldsymbol{x} = \left[\boldsymbol{h}_i \| \boldsymbol{h}_{\varphi(i)}\right] \\ 0 & \text{otherwise} \end{cases} \tag{8}$$

Next, we define a *continues* function $\widetilde{g} \colon \mathbb{R}^{2d} \to \mathbb{R}$ that equals to $g$ in only specific $n^2$ inputs:

$$\widetilde{g}([\boldsymbol{h}_i \| \boldsymbol{h}_j]) = g([\boldsymbol{h}_i \| \boldsymbol{h}_j]), \forall i, j \in [n] \tag{9}$$

---
[11]https://grlmila.github.io/

For all other inputs $x \in \mathbb{R}^{2d}$, $\widetilde{g}(x)$ realizes to any values that maintain the continuity of $\widetilde{g}$ (this is possible because we fixed the values of $\widetilde{g}$ for only a finite set of points). [12]

Thus, for every node $i \in \mathcal{V}$ and $j_{\neq \varphi(i)} \in \mathcal{V}$:

$$1 = \widetilde{g}\left(\left[\boldsymbol{h}_i \| \boldsymbol{h}_{\varphi(i)}\right]\right) > \widetilde{g}\left(\left[\boldsymbol{h}_i \| \boldsymbol{h}_j\right]\right) = 0 \tag{10}$$

If we concatenate the two input vectors, and define the scoring function $e$ of GATv2 (Equation (7)) as a function of the concatenated vector $[\boldsymbol{h}_i \| \boldsymbol{h}_j]$, from the universal approximation theorem (Hornik et al., 1989; Cybenko, 1989; Funahashi, 1989; Hornik, 1991), $e$ can approximate $\widetilde{g}$ for any compact subset of $\mathbb{R}^{2d}$.

Thus, for any sufficiently small $\epsilon$ (any $0 < \epsilon < 1/2$) there exist parameters $\boldsymbol{W}$ and $\boldsymbol{a}$ such that for every node $i \in \mathcal{V}$ and every $j_{\neq \varphi(i)}$:

$$e_{\boldsymbol{W},\boldsymbol{a}}\left(\boldsymbol{h}_i, \boldsymbol{h}_{\varphi(i)}\right) > 1 - \epsilon > 0 + \epsilon > e_{\boldsymbol{W},\boldsymbol{a}}\left(\boldsymbol{h}_i, \boldsymbol{h}_j\right) \tag{11}$$

and due to the increasing monotonicity of $\mathrm{softmax}$:

$$\alpha_{i,\varphi(i)} > \alpha_{i,j} \tag{12}$$

$\square$

**The choice of nonlinearity**  In general, these results hold if GATv2 had used any common non-polynomial activation function (such as ReLU, sigmoid, or the hyperbolic tangent function). The LeakyReLU activation function of GATv2 does not change its universal approximation ability (Leshno et al., 1993; Pinkus, 1999; Park et al., 2021), and it was chosen only for consistency with the original definition of GAT.

## C  GENERALIZATION TO MULTI-HEAD ATTENTION

Veličković et al. (2018) found it beneficial to employ $H$ separate attention heads and concatenate their outputs, similarly to Transformers. In this case, Theorem 1 holds for each head separately: every head $h \in [H]$ has a (possibly different) node that maximizes $\{s_j^{(h)} \mid j \in \mathcal{V}\}$, and the output is the concatenation of $H$ static attention heads.

## D  TRAINING DETAILS

In this section we elaborate on the training details of all of our experiments. All models use residual connections as in Veličković et al. (2018). All used code and data are publicly available under the MIT license.

### D.1  NODE- AND LINK-PREDICTION

We used the provided splits of OGB (Hu et al., 2020) and the Adam optimizer. We tuned the following hyperparameters: number of layers $\in \{2, 3, 6\}$, hidden size $\in \{64, 128, 256\}$, learning rate $\in \{0.0005, 0.001, 0.005, 0.01\}$ and sampling method – full batch, GraphSAINT (Zeng et al., 2019) and NeighborSampling (Hamilton et al., 2017). We tuned hyperparameters according to validation score and early stopping. The final hyperparameters are detailed in Table 4.

---

[12]The function $\widetilde{g}$ is a function that we define for the ease of proof, because the universal approximation theorem is defined for continuous functions, and we only need the scoring function of GATv2 $e$ to approximate the mapping $\varphi$ in a finite set of points. So, we need the attention function $e$ to approximate $g$ (from Equation 8) in some specific points. But, since $g$ is not continuous, we define $\widetilde{g}$ and use the universal approximation theorem for $\widetilde{g}$. Since $e$ approximates $\widetilde{g}$, $e$ also approximates $g$ in our specific points, as a special case. We only require that $\widetilde{g}$ will be identical to $g$ in specific $n^2$ points $\{[h_i \| h_j] \mid i, j \in [n]\}$. For the rest of the input space, we don't have any requirement on the value of $\widetilde{g}$, except for maintaining the continuity of $\widetilde{g}$. There exist infinitely many such possible $\widetilde{g}$ for every given set of keys, queries and a mapping $\varphi$, but the concrete functions are not needed for the proof.

| Dataset | # layers | Hidden size | Learning rate | Sampling method |
|---------|----------|-------------|---------------|-----------------|
| **ogbn-arxiv** | 3 | 256 | 0.01 | GraphSAINT |
| **ogbn-products** | 3 | 128 | 0.001 | NeighborSampling |
| **ogbn-mag** | 2 | 256 | 0.01 | NeighborSampling |
| **ogbn-proteins** | 6 | 64 | 0.01 | NeighborSampling |
| **ogbl-collab** | 3 | 64 | 0.001 | Full Batch |
| **ogbl-citation2** | 3 | 256 | 0.0005 | NeighborSampling |

Table 4: Training details of node- and link-prediction datasets.

## D.2 ROBUSTNESS TO NOISE

In these experiments, we used the same best-found hyperparameters in node-prediction, with 8 attention heads in **ogbn-arxiv** and 1 head in **ogbn-mag**. Each point is an average of 10 runs.

## D.3 SYNTHETIC BENCHMARK: DICTIONARYLOOKUP

In all experiments, we used a learning rate decay of $0.5$, a hidden size of $d = 128$, a batch size of $1024$, and the Adam optimizer.

We created a separate dataset for every graph size ($k$), and we split each such dataset to train and test with a ratio of 80:20. Since this is a contrived problem, we did not use a validation set, and the reported test results can be thought of as validation results. Every model was trained on a fixed value of $k$. Every key node (bottom row in Figure 2) was encoded as a sum of learned attribute embedding and a value embedding, followed by ReLU.

We experimented with layer normalization, batch normalization, dropout, various activation functions and various learning rates. None of these changed the general trend, so the experiments in Figure 3 were conducted without any normalization, without dropout and a learning rate of $0.001$.

## D.4 PROGRAMS: VARMISUSE

We used the code, splits, and the same best-found configurations as Brockschmidt (2020), who performed an extensive hyperparameter tuning by searching over 30 configurations for each GNN type. We trained each model five times.

We took the best-found hyperparameters of Brockschmidt (2020) for GAT and used them to train GATv2, without any further tuning.

## D.5 GRAPH-PREDICTION: QM9

We used the code and splits of Brockschmidt (2020) who performed an extensive hyperparameter search over 500 configurations. We took the best-found hyperparameters of Brockschmidt (2020) for GAT and used them to train GATv2. The only minor change from GAT is placing a residual connection after every layer, rather than after every other layer, which is within the experimented hyperparameter search that was reported by Brockschmidt (2020).

## D.6 COMPUTE AND RESOURCES

Our experiments consumed approximately 100 days of GPU in total. We used cloud GPUs of type V100, and we used RTX 3080 and 3090 in local GPU machines.

# E DATA STATISTICS

## E.1 NODE- AND LINK-PREDICTION DATASETS

Statistics of the OGB datasets we used for node- and link-prediction are shown in Table 5.

| Dataset | # nodes | # edges | Avg. node degree | Diameter |
|---|---|---|---|---|
| **ogbn-arxiv** | 169,343 | 1,166,243 | 13.7 | 23 |
| **ogbn-mag** | 1,939,743 | 21,111,007 | 21.7 | 6 |
| **ogbn-products** | 2,449,029 | 61,859,140 | 50.5 | 27 |
| **ogbn-proteins** | 132,534 | 39,561,252 | 597.0 | 9 |
| **ogbl-collab** | 235,868 | 1,285,465 | 8.2 | 22 |
| **ogbl-citation2** | 2,927,963 | 30,561,187 | 20.7 | 21 |

Table 5: Statistics of the OGB datasets (Hu et al., 2020).

### E.2 QM9

Statistics of the QM9 dataset, as used in Brockschmidt (2020) are shown in Table 6.

| | Training | Validation | Test |
|---|---|---|---|
| # examples | 110,462 | 10,000 | 10,000 |
| # nodes - average | 18.03 | 18.06 | 18.09 |
| # edges - average | 18.65 | 18.67 | 18.72 |
| Diameter - average | 6.35 | 6.35 | 6.35 |

Table 6: Statistics of the QM9 chemical dataset (Ramakrishnan et al., 2014) as used by Brockschmidt (2020).

### E.3 VARMISUSE

Statistics of the VARMISUSE dataset, as used in Allamanis et al. (2018) and Brockschmidt (2020), are shown in Table 7.

| | Training | Validation | UnseenProject Test | SeenProject Test |
|---|---|---|---|---|
| # graphs | 254360 | 42654 | 117036 | 59974 |
| # nodes - average | 2377 | 1742 | 1959 | 3986 |
| # edges - average | 7298 | 7851 | 5882 | 12925 |
| Diameter - average | 7.88 | 7.88 | 7.78 | 7.82 |

Table 7: Statistics of the VARMISUSE dataset (Allamanis et al., 2018) as used by Brockschmidt (2020).

## F ADDITIONAL RESULTS

### F.1 DICTIONARYLOOKUP

Figure 5 shows additional comparison between GATv2 and GIN (Xu et al., 2019) in the DICTIO-NARYLOOKUP problem. GATv2 easily achieves 100% train and test accuracy even for $k=100$ and using only a single head. GIN, although considered as more expressive than other GNNs, cannot perfectly fit the training data (with a model size of $d = 128$) starting from $k=20$.

### F.2 LINK-PREDICTION

Results for link-prediction are showen in Table 8.

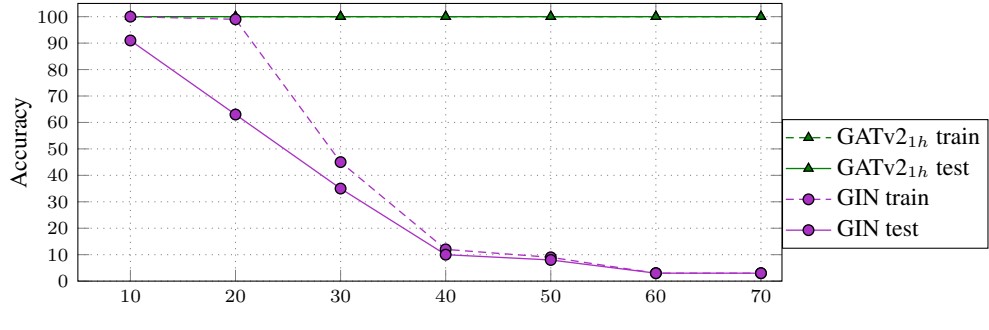

Figure 5: Train and test accuracy across graph sizes in the DICTIONARYLOOKUP problem. GATv2 easily achieves 100% train and test accuracy even for $k{=}100$ and using only a single head. GIN (Xu et al., 2019), although considered as more expressive than other GNNs, cannot perfectly fit the training data (with a model size of $d = 128$) starting from $k{=}20$.

Table 8: Average Hits@50 (Table 8a) and mean reciprocal rank (MRR) (Table 8b) in link-prediction benchmarks from OGB (10 runs±std). The best result among GAT and GATv2 is marked in **bold**; the best result among all GNNs is marked in **bold and underline**. † was reported by Hu et al. (2020).

| | | (a) | | (b) |
| | | **ogbl-collab** | | **ogbl-citation2** |
| Model | Attn. Heads | w/o val edges | w/ val edges | |
| No- | GCN† | $44.75_{\pm1.07}$ | $47.14_{\pm1.45}$ | $80.04_{\pm0.25}$ |
| Attention | GraphSAGE† | **$48.10_{\pm0.81}$** | **$54.63_{\pm1.12}$** | **$80.44_{\pm0.10}$** |
| GAT | $GAT_{1h}$ | $39.32_{\pm3.26}$ | $48.10_{\pm4.80}$ | $79.84_{\pm0.19}$ |
| | $GAT_{8h}$ | $42.37_{\pm2.99}$ | $46.63_{\pm2.80}$ | $75.95_{\pm1.31}$ |
| GATv2 | $GATv2_{1h}$ | $42.00_{\pm2.40}$ | $48.02_{\pm2.77}$ | $80.33_{\pm0.13}$ |
| | $GATv2_{8h}$ | **$42.85_{\pm2.64}$** | **$49.70_{\pm3.08}$** | **$80.14_{\pm0.71}$** |

### F.3 QM9

Standard deviation for the QM9 results of Section 4.5 are presented in Table 9.

### F.4 PUBMED CITATION NETWORK

We tuned the following parameters for both GAT and GATv2: number of layers $\in \{0, 1, 2\}$, hidden size $\in \{8, 16, 32\}$, number of heads $\in \{1, 4, 8\}$, dropout $\in \{0.4, 0.6, 0.8\}$, bias $\in \{True, False\}$, share weights $\in \{True, False\}$, use residual $\in \{True, False\}$. Table 10 shows the test accuracy (100 runs±stdev) using the best hyperparameters found for each model.

It is important to note that PubMed has only **60 training nodes**, which hinders expressive models such as GATv2 from exploiting their approximation and generalization advantages. Still, GATv2 is more accurate than GAT even in this small dataset. In Table 15, we show that this difference is statistically significant (p-value $< 0.0001$).

## G  ADDITIONAL COMPARISON WITH TRANSFORMER-STYLE ATTENTION (DPGAT)

The main goal of our paper is to highlight a severe theoretical limitation of the highly popular GAT architecture, and propose a minimal fix.

| Model | Predicted Property | | | | | | |
|---|---|---|---|---|---|---|---|
| | 1 | 2 | 3 | 4 | 5 | 6 | 7 |
| GCN[†] | $3.21_{\pm 0.06}$ | $\mathbf{4.22}_{\pm 0.45}$ | $1.45_{\pm 0.01}$ | $1.62_{\pm 0.04}$ | $2.42_{\pm 0.14}$ | $16.38_{\pm 0.49}$ | $17.40_{\pm 3.56}$ |
| GIN[†] | $\mathbf{2.64}_{\pm 0.11}$ | $4.67_{\pm 0.52}$ | $1.42_{\pm 0.01}$ | $1.50_{\pm 0.09}$ | $\mathbf{2.27}_{\pm 0.09}$ | $\underline{\mathbf{15.63}}_{\pm 1.40}$ | $\mathbf{12.93}_{\pm 1.81}$ |
| GAT$_{1h}$ | $3.08_{\pm 0.08}$ | $7.82_{\pm 1.42}$ | $1.79_{\pm 0.10}$ | $3.96_{\pm 1.51}$ | $3.58_{\pm 1.03}$ | $35.43_{\pm 29.9}$ | $116.5_{\pm 10.65}$ |
| GAT$_{8h}$[†] | $2.68_{\pm 0.06}$ | $4.65_{\pm 0.44}$ | $1.48_{\pm 0.03}$ | $1.53_{\pm 0.07}$ | $2.31_{\pm 0.06}$ | $52.39_{\pm 42.58}$ | $14.87_{\pm 2.88}$ |
| GATv2$_{1h}$ | $3.04_{\pm 0.06}$ | $6.38_{\pm 0.62}$ | $1.68_{\pm 0.04}$ | $2.18_{\pm 0.61}$ | $2.82_{\pm 0.25}$ | $20.56_{\pm 0.70}$ | $77.13_{\pm 37.93}$ |
| GATv2$_{8h}$ | $\mathbf{2.65}_{\pm 0.05}$ | $\mathbf{4.28}_{\pm 0.27}$ | $\underline{\mathbf{1.41}}_{\pm 0.04}$ | $\underline{\mathbf{1.47}}_{\pm 0.03}$ | $2.29_{\pm 0.15}$ | $\mathbf{16.37}_{\pm 0.97}$ | $\mathbf{14.03}_{\pm 1.39}$ |

| Model | Predicted Property | | | | | | Rel. to GAT$_{8h}$ |
|---|---|---|---|---|---|---|---|
| | 8 | 9 | 10 | 11 | 12 | 13 | |
| GCN[†] | $7.82_{\pm 0.80}$ | $8.24_{\pm 1.25}$ | $9.05_{\pm 1.21}$ | $7.00_{\pm 1.51}$ | $3.93_{\pm 0.48}$ | $\underline{\mathbf{1.02}}_{\pm 0.05}$ | -1.5% |
| GIN[†] | $\mathbf{5.88}_{\pm 1.01}$ | $18.71_{\pm 23.36}$ | $\mathbf{5.62}_{\pm 0.81}$ | $\mathbf{5.38}_{\pm 0.75}$ | $\mathbf{3.53}_{\pm 0.37}$ | $1.05_{\pm 0.11}$ | -2.3% |
| GAT$_{1h}$ | $28.10_{\pm 16.45}$ | $20.80_{\pm 13.40}$ | $15.80_{\pm 5.87}$ | $10.80_{\pm 2.18}$ | $5.37_{\pm 0.26}$ | $3.11_{\pm 0.14}$ | +134.1% |
| GAT$_{8h}$[†] | $7.61_{\pm 0.46}$ | $6.86_{\pm 0.53}$ | $7.64_{\pm 0.92}$ | $6.54_{\pm 0.36}$ | $4.11_{\pm 0.27}$ | $1.48_{\pm 0.87}$ | +0% |
| GATv2$_{1h}$ | $10.19_{\pm 0.63}$ | $22.56_{\pm 17.46}$ | $15.04_{\pm 4.58}$ | $22.94_{\pm 17.34}$ | $5.23_{\pm 0.36}$ | $2.46_{\pm 0.65}$ | +91.6% |
| GATv2$_{8h}$ | $\mathbf{6.07}_{\pm 0.77}$ | $\underline{\mathbf{6.28}}_{\pm 0.83}$ | $\mathbf{6.60}_{\pm 0.79}$ | $\mathbf{5.97}_{\pm 0.94}$ | $3.57_{\pm 0.36}$ | $1.59_{\pm 0.96}$ | $\underline{\mathbf{-11.5}}$% |

Table 9: Average error rates (lower is better), 5 runs $\pm$ standard deviation for each property, on the QM9 dataset. The best result among GAT and GATv2 is marked in **bold**; the globally best result among all GNNs is marked in **bold and underline**. † was previously tuned and reported by Brockschmidt (2020).

Table 10: Accuracy (100 runs±stdev) on Pubmed. GATv2 is more accurate than GAT.

| Model | Accuracy |
|---|---|
| GAT | $78.1_{\pm 0.59}$ |
| GATv2 | $\mathbf{78.5}_{\pm 0.38}$ |

We perform additional empirical comparison to DPGAT, which follows Luong et al. (2015) and the dot-product attention of the Transformer (Vaswani et al., 2017). We define DPGAT as:

$$\text{DPGAT (Vaswani et al., 2017):} \qquad e\left(\boldsymbol{h}_i, \boldsymbol{h}_j\right) = \left(\left(\boldsymbol{h}_i^\top \boldsymbol{Q}\right) \cdot \left(\boldsymbol{h}_j^\top \boldsymbol{K}\right)^\top\right) / \sqrt{d_k} \qquad (13)$$

Variants of DPGAT were used in prior work (Gao and Ji, 2019; Dwivedi and Bresson, 2020; Rong et al., 2020a; Veličković et al., 2020; Kim and Oh, 2021), and we consider it here for the conceptual and empirical comparison with GAT.

Despite its popularity, DPGAT is *strictly weaker* than GATv2. DPGAT provably performs dynamic attention for any set of node representations only if they are *linearly independent* (see Theorem 3 and its proof in Appendix G.1). Otherwise, there are examples of node representations that *are* linearly dependent and mappings $\varphi$, for which dynamic attention does not hold (Appendix G.2). This constraint is not harmful when violated in practice, because every node has only a small set of neighbors, rather than all possible nodes in the graph; further, some nodes possibly never need to be "selected" in practice.

### G.1 PROOF THAT DPGAT PERFORMS DYNAMIC ATTENTION FOR LINEARLY INDEPENDENT NODE REPRESENTATIONS

**Theorem 3.** *A DPGAT layer computes dynamic attention for any set of node representations* $\mathbb{K} = \mathbb{Q} = \{\boldsymbol{h}_1, ..., \boldsymbol{h}_n\}$ *that are linearly independent.*

*Proof.* Let $\mathcal{G} = (\mathcal{V}, \mathcal{E})$ be a graph modeled by a DPGAT layer, having linearly independent node representations $\{\boldsymbol{h}_1, ..., \boldsymbol{h}_n\}$. Let $\varphi : [n] \to [n]$ be any node mapping $[n] \to [n]$.

We denote the $\text{i}^{th}$ row of a matrix $M$ as $M_i$.

We define a matrix $P$ as:

$$P_{i,j} = \begin{cases} 1 & j = \varphi(i) \\ 0 & \text{otherwise} \end{cases} \tag{14}$$

Let $X \in \mathbb{R}^n \times \mathbb{R}^d$ be the matrix holding the graph's node representations as its rows:

$$X = \begin{bmatrix} — & h_1 & — \\ — & h_2 & — \\ & \vdots & \\ — & h_n & — \end{bmatrix} \tag{15}$$

Since the rows of $X$ are linearly independent, it necessarily holds that $d \geq n$.

Next, we find weight matrices $Q \in \mathbb{R}^d \times \mathbb{R}^d$ and $K \in \mathbb{R}^d \times \mathbb{R}^d$ such that:

$$(XQ) \cdot (XK)^\top = P \tag{16}$$

To satisfy Equation (16), we choose $Q$ and $K$ such that $XQ = U$ and $XK = P^\top U$ where $U$ is an orthonormal matrix ($U \cdot U^\top = U^\top \cdot U = I$).

We can obtain $U$ using the singular value decomposition (SVD) of $X$:

$$X = U\Sigma V^\top \tag{17}$$

Since $\Sigma \in \mathbb{R}^n \times \mathbb{R}^n$ and $X$ has a full rank, $\Sigma$ is invertible, and thus:

$$XV\Sigma^{-1} = U \tag{18}$$

Now, we define $Q$ as follows:

$$Q = V\Sigma^{-1} \tag{19}$$

Note that $XQ = U$, as desired.

To find $K$ that satisfies $XK = P^\top U$, we use Equation (17) and require:

$$U\Sigma V^\top K = P^\top U \tag{20}$$

and thus:

$$K = V\Sigma^{-1} U^T P^\top U \tag{21}$$

We define:

$$z(h_i, h_j) = e(h_i, h_j) \cdot \sqrt{d_k} \tag{22}$$

Where $e$ is the attention score function of DPGAT (Equation (13)).

Now, for a query $i$ and a key $j$, and the corresponding representations $h_i, h_j$:

$$z(h_i, h_j) = \left(h_i^\top Q\right) \cdot \left(h_j^\top K\right)^\top \tag{23}$$

$$= (X_i Q) \cdot (X_j K)^\top \tag{24}$$

Since $X_i Q = (XQ)_i$ and $X_j K = (XK)_j$, we get

$$z(h_i, h_j) = (XQ)_i \cdot \left((XK)_j\right)^\top = P_{i,j} \tag{25}$$

Therefore:

$$z(h_i, h_j) = \begin{cases} 1 & j = \varphi(i) \\ 0 & \text{otherwise} \end{cases} \tag{26}$$

And thus:

$$e(h_i, h_j) = \begin{cases} 1/\sqrt{d_k} & j = \varphi(i) \\ 0 & otherwise \end{cases} \tag{27}$$

To conclude, for every selected query $i$ and any key $j_{\neq\varphi(i)}$:

$$e\left(\boldsymbol{h}_i, \boldsymbol{h}_{\varphi(i)}\right) > e\left(\boldsymbol{h}_i, \boldsymbol{h}_j\right) \tag{28}$$

and due to the increasing monotonicity of $\mathrm{softmax}$:

$$\alpha_{i,\varphi(i)} > \alpha_{i,j} \tag{29}$$

Hence, a DPGAT layer computes dynamic attention.

In the case that $d > n$, we apply SVD to the full-rank matrix $\boldsymbol{X}\boldsymbol{X}^\top \in \mathbb{R}^{n \times n}$, and follow the same steps to construct $\boldsymbol{Q}$ and $\boldsymbol{K}$.

In the case that $\boldsymbol{Q} \in \mathbb{R}^d \times \mathbb{R}^{d_k}$ and $\boldsymbol{K} \in \mathbb{R}^d \times \mathbb{R}^{d_k}$ and $d_k > d$, we can use the same $\boldsymbol{Q}$ and $\boldsymbol{K}$ (Equations (19) and (21)) padded with zeros. We define the $\boldsymbol{Q}' \in \mathbb{R}^d \times \mathbb{R}^{d_{key}}$ and $\boldsymbol{K}' \in \mathbb{R}^d \times \mathbb{R}^{d_{key}}$ as follows:

$$\boldsymbol{Q}'_{i,j} = \begin{cases} \boldsymbol{Q}_{i,j} & j \le d \\ 0 & \text{otherwise} \end{cases} \tag{30}$$

$$\boldsymbol{K}'_{i,j} = \begin{cases} \boldsymbol{K}_{i,j} & j \le d \\ 0 & \text{otherwise} \end{cases} \tag{31}$$

$\square$

## G.2 DPGAT IS STRICTLY WEAKER THAN GATV2

There are examples of node representations that are linearly dependent and mappings $\varphi$, for which dynamic attention does not hold. First, we show a simple 2-dimensional example, and then we show the general case of such examples.

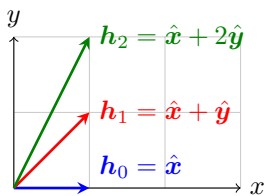

Figure 6: An example for node representations that are linearly dependent, for which DPGAT cannot compute dynamic attention, because no query vector $\boldsymbol{q} \in \mathbb{R}^2$ can "select" $\boldsymbol{h}_1$.

Consider the following linearly dependent set of vectors $\mathbb{K} = \mathbb{Q}$ (Figure 6):

$$\boldsymbol{h}_0 = \hat{\boldsymbol{x}} \tag{32}$$
$$\boldsymbol{h}_1 = \hat{\boldsymbol{x}} + \hat{\boldsymbol{y}} \tag{33}$$
$$\boldsymbol{h}_2 = \hat{\boldsymbol{x}} + 2\hat{\boldsymbol{y}} \tag{34}$$

where $\hat{\boldsymbol{x}}$ and $\hat{\boldsymbol{y}}$ are the cartesian unit vectors. We define $\beta \in \{0, 1, 2\}$ to express $\{\boldsymbol{h}_0, \boldsymbol{h}_1, \boldsymbol{h}_2\}$ using the same expression:

$$\boldsymbol{h}_\beta = \hat{\boldsymbol{x}} + \beta\hat{\boldsymbol{y}} \tag{35}$$

Let $\boldsymbol{q} \in \mathbb{Q}$ be any query vector. For brevity, we define the unscaled dot-product attention score as $s$:

$$s\left(\boldsymbol{q}, \boldsymbol{h}_\beta\right) = e\left(\boldsymbol{q}, \boldsymbol{h}_\beta\right) \cdot \sqrt{d_k} \tag{36}$$

Where $e$ is the attention score function of DPGAT (Equation (13)). The (unscaled) attention score between $\boldsymbol{q}$ and $\{\boldsymbol{h}_0, \boldsymbol{h}_1, \boldsymbol{h}_2\}$ is:

$$s\left(\boldsymbol{q}, \boldsymbol{h}_\beta\right) = \left(\boldsymbol{q}^\top \boldsymbol{Q}\right)\left(\boldsymbol{h}_\beta^\top \boldsymbol{K}\right)^\top \tag{37}$$

$$= \left(\boldsymbol{q}^\top \boldsymbol{Q}\right)\left(\left(\hat{\boldsymbol{x}} + \beta\hat{\boldsymbol{y}}\right)^\top \boldsymbol{K}\right)^\top \tag{38}$$

$$= \left(\boldsymbol{q}^\top \boldsymbol{Q}\right)\left(\hat{\boldsymbol{x}}^\top \boldsymbol{K} + \beta\hat{\boldsymbol{y}}^\top \boldsymbol{K}\right)^\top \tag{39}$$

$$= \left(\boldsymbol{q}^\top \boldsymbol{Q}\right)\left(\hat{\boldsymbol{x}}^\top \boldsymbol{K}\right)^\top + \beta\left(\boldsymbol{q}^\top \boldsymbol{Q}\right)\left(\hat{\boldsymbol{y}}^\top \boldsymbol{K}\right)^\top \tag{40}$$

The first term $\left(\boldsymbol{q}^\top \boldsymbol{Q}\right)\left(\hat{\boldsymbol{x}}^\top \boldsymbol{K}\right)^\top$ is unconditioned on $\beta$, and thus shared for every $\boldsymbol{h}_\beta$. Let us focus on the second term $\beta\left(\boldsymbol{q}^\top \boldsymbol{Q}\right)\left(\hat{\boldsymbol{y}}^\top \boldsymbol{K}\right)^\top$. If $\left(\boldsymbol{q}^\top \boldsymbol{Q}\right)\left(\hat{\boldsymbol{y}}^\top \boldsymbol{K}\right)^\top > 0$, then:

$$e\left(\boldsymbol{q}, \boldsymbol{h}_2\right) > e\left(\boldsymbol{q}, \boldsymbol{h}_1\right) \tag{41}$$

Otherwise, if $\left(\boldsymbol{q}^\top \boldsymbol{Q}\right)\left(\hat{\boldsymbol{y}}^\top \boldsymbol{K}\right)^\top \leq 0$:

$$e\left(\boldsymbol{q}, \boldsymbol{h}_0\right) \geq e\left(\boldsymbol{q}, \boldsymbol{h}_1\right) \tag{42}$$

Thus, for any query $\boldsymbol{q}$, the key $\boldsymbol{h}_1$ can never get the highest score, and thus cannot be "selected". That is, the key $\boldsymbol{h}_1$ cannot satisfy that $e\left(\boldsymbol{q}, \boldsymbol{h}_1\right)$ is strictly greater than any other key.

In the general case, let $\boldsymbol{h}_0, \boldsymbol{h}_1 \in \mathbb{R}^d$ be some non-zero vectors , and $\lambda$ is some scalar such that $0 < \lambda < 1$.

Consider the following linearly dependent set of vectors:

$$\mathbb{K} = \mathbb{Q} = \{\beta \boldsymbol{h}_1 + (1 - \beta) \boldsymbol{h}_0 \mid \beta \in \{0, \lambda, 1\}\} \tag{43}$$

For any query $\boldsymbol{q} \in \mathbb{Q}$ and $\beta \in \{0, \lambda, 1\}$ we define:

$$s\left(\boldsymbol{q}, \beta\right) = e\left(\boldsymbol{q}, \left(\beta \boldsymbol{h}_1 + (1 - \beta) \boldsymbol{h}_0\right)\right) \cdot \sqrt{d_k} \tag{44}$$

Where $e$ is the attention score function of DPGAT (Equation (13)).

Therefore:

$$s\left(\boldsymbol{q}, \beta\right) = \left(\boldsymbol{q}^\top \boldsymbol{Q}\right)\left(\left(\beta \boldsymbol{h}_1 + (1 - \beta) \boldsymbol{h}_0\right)^\top \boldsymbol{K}\right)^\top \tag{45}$$

$$= \left(\boldsymbol{q}^\top \boldsymbol{Q}\right)\left(\beta \boldsymbol{h}_1^\top \boldsymbol{K} + (1 - \beta) \boldsymbol{h}_0^\top \boldsymbol{K}\right)^\top \tag{46}$$

$$= \left(\boldsymbol{q}^\top \boldsymbol{Q}\right)\left(\beta \boldsymbol{h}_1^\top \boldsymbol{K} + \boldsymbol{h}_0^\top \boldsymbol{K} - \beta \boldsymbol{h}_0^\top \boldsymbol{K}\right)^\top \tag{47}$$

$$= \left(\boldsymbol{q}^\top \boldsymbol{Q}\right)\left(\beta \left(\boldsymbol{h}_1^\top \boldsymbol{K} - \boldsymbol{h}_0^\top \boldsymbol{K}\right) + \boldsymbol{h}_0^\top \boldsymbol{K}\right)^\top \tag{48}$$

$$= \beta \left(\boldsymbol{q}^\top \boldsymbol{Q}\right)\left(\boldsymbol{h}_1^\top \boldsymbol{K} - \boldsymbol{h}_0^\top \boldsymbol{K}\right)^\top + \left(\boldsymbol{q}^\top \boldsymbol{Q}\right)\left(\boldsymbol{h}_0^\top \boldsymbol{K}\right)^\top \tag{49}$$

If $\left(\boldsymbol{q}^\top \boldsymbol{Q}\right)\left(\boldsymbol{h}_1^\top \boldsymbol{K} - \boldsymbol{h}_0^\top \boldsymbol{K}\right)^\top > 0$:

$$e\left(\boldsymbol{q}, \boldsymbol{h}_1\right) > e\left(\boldsymbol{q}, \boldsymbol{h}_\lambda\right) \tag{50}$$

Otherwise, if $\left(\boldsymbol{q}^\top \boldsymbol{Q}\right)\left(\boldsymbol{h}_1^\top \boldsymbol{K} - \boldsymbol{h}_0^\top \boldsymbol{K}\right)^\top \leq 0$:

$$e\left(\boldsymbol{q}, \boldsymbol{h}_0\right) \geq e\left(\boldsymbol{q}, \boldsymbol{h}_\lambda\right) \tag{51}$$

Thus, for any query $\boldsymbol{q}$, the key $\boldsymbol{h}_\lambda$ cannot be selected. That is, the key $\boldsymbol{h}_\lambda$ cannot satisfy that $e\left(\boldsymbol{q}, \boldsymbol{h}_\lambda\right)$ is strictly greater than any other key. Therefore, there are mappings $\varphi$, for which dynamic attention does not hold.

While we prove that GATv2 computes dynamic attention (Appendix B) for *any* set of node representations $\mathbb{K} = \mathbb{Q}$, there are sets of node representations and mappings $\varphi$ for which dynamic attention does not hold for DPGAT. Thus, DPGAT is strictly weaker than GATv2.

### G.3  EMPIRICAL EVALUATION

Here we repeat the experiments of Section 4 with DPGAT. We remind that DPGAT is *strictly weaker* than our proposed GATv2 (see a proof in Appendix G.1).

## H  STATISTICAL SIGNIFICANCE

Here we report the statistical significance of the strongest GATv2 and GAT models of the experiments reported in Section 4.

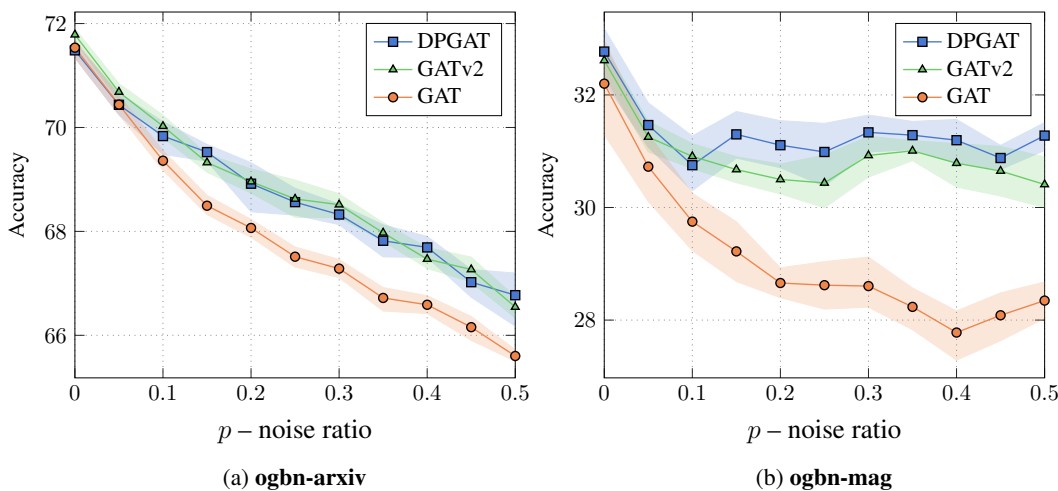

Figure 7: Test accuracy compared to the noise ratio: GATv2 and DPGAT are more robust to structural noise compared to GAT. Each point is an average of 10 runs, error bars show standard deviation.

Table 11: Accuracy (5 runs±stdev) on VARMISUSE. GATv2 is more accurate than all GNNs in both test sets, using GAT's hyperparameters. † – previously reported by Brockschmidt (2020).

|  | Model | SeenProj | UnseenProj |
|---|---|---|---|
| No-Attention | GCN† | $87.2_{\pm 1.5}$ | $81.4_{\pm 2.3}$ |
|  | GIN† | $87.1_{\pm 0.1}$ | $81.1_{\pm 0.9}$ |
| Attention | GAT† | $86.9_{\pm 0.7}$ | $81.2_{\pm 0.9}$ |
|  | DPGAT | $\mathbf{88.0}_{\pm 0.8}$ | $81.5_{\pm 1.2}$ |
|  | GATv2 | $\mathbf{88.0}_{\pm 1.1}$ | $\mathbf{82.8}_{\pm 1.7}$ |

Table 12: Average accuracy (Table 12a) and ROC-AUC (Table 12b) in node-prediction datasets (10 runs±std). In all datasets, GATv2 outperforms GAT. † – previously reported by Hu et al. (2020).

|  |  | (a) |  |  | (b) |
|---|---|---|---|---|---|
| Model | Attn. Heads | **ogbn-arxiv** | **ogbn-products** | **ogbn-mag** | **ogbn-proteins** |
| GCN† | 0 | $71.74_{\pm 0.29}$ | $78.97_{\pm 0.33}$ | $30.43_{\pm 0.25}$ | $72.51_{\pm 0.35}$ |
| GraphSAGE† | 0 | $71.49_{\pm 0.27}$ | $78.70_{\pm 0.36}$ | $31.53_{\pm 0.15}$ | $77.68_{\pm 0.20}$ |
| GAT | 1 | $71.59_{\pm 0.38}$ | $79.04_{\pm 1.54}$ | $32.20_{\pm 1.46}$ | $70.77_{\pm 5.79}$ |
|  | 8 | $71.54_{\pm 0.30}$ | $77.23_{\pm 2.37}$ | $31.75_{\pm 1.60}$ | $78.63_{\pm 1.62}$ |
| DPGAT | 1 | $71.52_{\pm 0.17}$ | $76.49_{\pm 0.78}$ | $\mathbf{32.77}_{\pm 0.80}$ | $63.47_{\pm 2.79}$ |
|  | 8 | $71.48_{\pm 0.26}$ | $73.53_{\pm 0.47}$ | $27.74_{\pm 9.97}$ | $72.88_{\pm 0.59}$ |
| GATv2 (this work) | 1 | $71.78_{\pm 0.18}$ | $\mathbf{80.63}_{\pm 0.70}$ | $32.61_{\pm 0.44}$ | $77.23_{\pm 3.32}$ |
|  | 8 | $\mathbf{71.87}_{\pm 0.25}$ | $78.46_{\pm 2.45}$ | $32.52_{\pm 0.39}$ | $\mathbf{79.52}_{\pm 0.55}$ |

Table 13: Average error rates (lower is better), 5 runs $\pm$ standard deviation for each property, on the QM9 dataset. The best result among GAT, GATv2 and DPGAT is marked in **bold**; the globally best result among all GNNs is marked in **bold and underline**. † was previously tuned and reported by Brockschmidt (2020).

| Model | Predicted Property | | | | | | |
|---|---|---|---|---|---|---|---|
| | 1 | 2 | 3 | 4 | 5 | 6 | 7 |
| GCN† | $3.21_{\pm0.06}$ | $\mathbf{4.22}_{\pm0.45}$ | $1.45_{\pm0.01}$ | $1.62_{\pm0.04}$ | $2.42_{\pm0.14}$ | $16.38_{\pm0.49}$ | $17.40_{\pm3.56}$ |
| GIN† | $2.64_{\pm0.11}$ | $4.67_{\pm0.52}$ | $1.42_{\pm0.01}$ | $1.50_{\pm0.09}$ | $2.27_{\pm0.09}$ | $\underline{15.63}_{\pm1.40}$ | $12.93_{\pm1.81}$ |
| GAT$_{1h}$ | $3.08_{\pm0.08}$ | $7.82_{\pm1.42}$ | $1.79_{\pm0.10}$ | $3.96_{\pm1.51}$ | $3.58_{\pm1.03}$ | $35.43_{\pm29.9}$ | $116.5_{\pm10.65}$ |
| GAT$_{8h}$† | $2.68_{\pm0.06}$ | $4.65_{\pm0.44}$ | $1.48_{\pm0.03}$ | $1.53_{\pm0.07}$ | $2.31_{\pm0.06}$ | $52.39_{\pm42.58}$ | $14.87_{\pm2.88}$ |
| DPGAT$_{8h}$ | $\underline{\mathbf{2.63}}_{\pm0.09}$ | $4.37_{\pm0.13}$ | $1.44_{\pm0.07}$ | $\underline{\mathbf{1.40}}_{\pm0.03}$ | $\underline{\mathbf{2.10}}_{\pm0.07}$ | $32.59_{\pm34.77}$ | $\underline{\mathbf{11.66}}_{\pm1.00}$ |
| DPGAT$_{1h}$ | $3.20_{\pm0.17}$ | $8.35_{\pm0.78}$ | $1.71_{\pm0.03}$ | $2.17_{\pm0.14}$ | $2.88_{\pm0.12}$ | $25.21_{\pm2.86}$ | $65.79_{\pm39.84}$ |
| GATv2$_{1h}$ | $3.04_{\pm0.06}$ | $6.38_{\pm0.62}$ | $1.68_{\pm0.04}$ | $2.18_{\pm0.61}$ | $2.82_{\pm0.25}$ | $20.56_{\pm0.70}$ | $77.13_{\pm37.93}$ |
| GATv2$_{8h}$ | $2.65_{\pm0.05}$ | $\mathbf{4.28}_{\pm0.27}$ | $\underline{\mathbf{1.41}}_{\pm0.04}$ | $1.47_{\pm0.03}$ | $2.29_{\pm0.15}$ | $16.37_{\pm0.97}$ | $14.03_{\pm1.39}$ |

| Model | Predicted Property | | | | | | Rel. to GAT$_{8h}$ |
|---|---|---|---|---|---|---|---|
| | 8 | 9 | 10 | 11 | 12 | 13 | |
| GCN† | $7.82_{\pm0.80}$ | $8.24_{\pm1.25}$ | $9.05_{\pm1.21}$ | $7.00_{\pm1.51}$ | $3.93_{\pm0.48}$ | $\underline{\mathbf{1.02}}_{\pm0.05}$ | -1.5% |
| GIN† | $\underline{\mathbf{5.88}}_{\pm1.01}$ | $18.71_{\pm23.36}$ | $\underline{\mathbf{5.62}}_{\pm0.81}$ | $\underline{\mathbf{5.38}}_{\pm0.75}$ | $\underline{\mathbf{3.53}}_{\pm0.37}$ | $1.05_{\pm0.11}$ | -2.3% |
| GAT$_{1h}$ | $28.10_{\pm16.45}$ | $20.80_{\pm13.40}$ | $15.80_{\pm5.87}$ | $10.80_{\pm2.18}$ | $5.37_{\pm0.26}$ | $3.11_{\pm0.14}$ | +134.1% |
| GAT$_{8h}$† | $7.61_{\pm0.46}$ | $6.86_{\pm0.53}$ | $7.64_{\pm0.92}$ | $6.54_{\pm0.36}$ | $4.11_{\pm0.27}$ | $1.48_{\pm0.87}$ | +0% |
| DPGAT$_{1h}$ | $12.93_{\pm1.70}$ | $13.32_{\pm2.39}$ | $14.42_{\pm1.95}$ | $13.83_{\pm2.55}$ | $6.37_{\pm0.28}$ | $3.28_{\pm1.16}$ | +77.9% |
| DPGAT$_{8h}$ | $6.95_{\pm0.32}$ | $7.09_{\pm0.59}$ | $7.30_{\pm0.66}$ | $6.52_{\pm0.61}$ | $3.76_{\pm0.21}$ | $\mathbf{1.18}_{\pm0.33}$ | -9.7% |
| GATv2$_{1h}$ | $10.19_{\pm0.63}$ | $22.56_{\pm17.46}$ | $15.04_{\pm4.58}$ | $22.94_{\pm17.34}$ | $5.23_{\pm0.36}$ | $2.46_{\pm0.65}$ | +91.6% |
| GATv2$_{8h}$ | $\mathbf{6.07}_{\pm0.77}$ | $\underline{\mathbf{6.28}}_{\pm0.83}$ | $\mathbf{6.60}_{\pm0.79}$ | $\mathbf{5.97}_{\pm0.94}$ | $3.57_{\pm0.36}$ | $1.59_{\pm0.96}$ | $\underline{\mathbf{-11.5}}$% |

Figure 8: Test accuracy and statistical significance compared to the noise ratio: GATv2 is more robust to structural noise compared to GAT. Each point is an average of 10 runs, error bars show standard deviation.

Table 14: Accuracy (5 runs$\pm$stdev) on VARMISUSE. GATv2 is more accurate than all GNNs in both test sets, using GAT's hyperparameters. † – previously reported by Brockschmidt (2020).

| Model | SeenProj | UnseenProj |
|---|---|---|
| GAT† | $86.9_{\pm0.7}$ | $81.2_{\pm0.9}$ |
| GATv2 | $\mathbf{88.0}_{\pm1.1}$ | $\mathbf{82.8}_{\pm1.7}$ |
| p-value | 0.048 | 0.049 |

Table 15: Accuracy (100 runs±stdev) on Pubmed. GATv2 is more accurate than GAT.

| Model | Accuracy |
|---|---|
| GAT | $78.1_{\pm 0.59}$ |
| GATv2 | $\mathbf{78.5}_{\pm 0.38}$ |
| p-value | < 0.0001 |

Table 16: Average accuracy (Table 16a) and ROC-AUC (Table 16b) in node-prediction datasets (30 runs±std). We report on the best GAT / GATv2 from Table 2.

(a)

| Model | ogbn-arxiv | ogbn-products | ogbn-mag |
|---|---|---|---|
| GAT | $71.65_{\pm 0.38}$ | $79.04_{\pm 1.54}$ | $32.36_{\pm 1.10}$ |
| GATv2 | $\mathbf{71.93}_{\pm 0.35}$ | $\mathbf{80.63}_{\pm 0.70}$ | $\mathbf{33.01}_{\pm 0.41}$ |
| p-value | 0.0022 | <0.0001 | 0.0018 |

(b)

| ogbn-proteins |
|---|
| $78.29_{\pm 1.59}$ |
| $\mathbf{78.96}_{\pm 1.19}$ |
| 0.0349 |

Table 17: Average Hits@50 (Table 17a) and mean reciprocal rank (MRR) (Table 17b) in link-prediction benchmarks from OGB (30 runs±std). We report on the best GAT / GATv2 from Table 8.

(a)

| Model | ogbl-collab | |
|---|---|---|
| | w/o val edges | w/ val edges |
| GAT | $42.24_{\pm 2.26}$ | $46.02_{\pm 4.09}$ |
| GATv2 | $\mathbf{43.82}_{\pm 2.24}$ | $\mathbf{49.06}_{\pm 2.50}$ |
| p-value | 0.0043 | 0.0005 |

(b)

| ogbl-citation2 |
|---|
| $79.91_{\pm 0.13}$ |
| $\mathbf{80.20}_{\pm 0.62}$ |
| 0.0075 |

Table 18: Average error rates (lower is better), 20 runs ± standard deviation for each property, on the QM9 dataset. We report on GAT and GATv2 with 8 attention heads.

| Model | Predicted Property | | | | | | |
|---|---|---|---|---|---|---|---|
| | 1 | 2 | 3 | 4 | 5 | 6 | 7 |
| GAT | $2.74_{\pm 0.08}$ | $4.73_{\pm 0.40}$ | $1.47_{\pm 0.06}$ | $1.53_{\pm 0.06}$ | $2.44_{\pm 0.60}$ | $55.21_{\pm 42.33}$ | $25.36_{\pm 31.42}$ |
| GATv2 | $\mathbf{2.67}_{\pm 0.08}$ | $\mathbf{4.28}_{\pm 0.23}$ | $\mathbf{1.43}_{\pm 0.05}$ | $\mathbf{1.51}_{\pm 0.07}$ | $\mathbf{2.21}_{\pm 0.08}$ | $\mathbf{16.64}_{\pm 1.17}$ | $\mathbf{13.61}_{\pm 1.68}$ |
| p-value | 0.0043 | <0.0001 | 0.0138 | 0.1691 | 0.0487 | 0.0001 | 0.0516 |

| Model | Predicted Property | | | | | |
|---|---|---|---|---|---|---|
| | 8 | 9 | 10 | 11 | 12 | 13 |
| GAT | $7.36_{\pm 0.87}$ | $6.79_{\pm 0.86}$ | $7.36_{\pm 0.93}$ | $6.69_{\pm 0.86}$ | $4.10_{\pm 0.29}$ | $1.51_{\pm 0.84}$ |
| GATv2 | $\mathbf{6.13}_{\pm 0.59}$ | $\mathbf{6.33}_{\pm 0.82}$ | $\mathbf{6.37}_{\pm 0.86}$ | $\mathbf{5.95}_{\pm 0.62}$ | $\mathbf{3.66}_{\pm 0.29}$ | $\mathbf{1.09}_{\pm 0.85}$ |
| p-value | <0.0001 | 0.0458 | 0.0006 | 0.0017 | <0.0001 | 0.0621 |

## I   COMPLEXITY ANALYSIS

We repeat the definitions of GAT, GATv2 and DPGAT:

$$\text{GAT (Veličković et al., 2018):} \qquad e\left(\boldsymbol{h}_i, \boldsymbol{h}_j\right) = \text{LeakyReLU}\left(\boldsymbol{a}^\top \cdot [\boldsymbol{W}\boldsymbol{h}_i \| \boldsymbol{W}\boldsymbol{h}_j]\right) \qquad (52)$$

$$\text{GATv2 (our fixed version):} \qquad e\left(\boldsymbol{h}_i, \boldsymbol{h}_j\right) = \boldsymbol{a}^\top \text{LeakyReLU}\left(\boldsymbol{W} \cdot [\boldsymbol{h}_i \| \boldsymbol{h}_j]\right) \qquad (53)$$

$$\text{DPGAT (Vaswani et al., 2017):} \qquad e\left(\boldsymbol{h}_i, \boldsymbol{h}_j\right) = \left(\left(\boldsymbol{h}_i^\top \boldsymbol{Q}\right) \cdot \left(\boldsymbol{h}_j^\top \boldsymbol{K}\right)^\top\right) / \sqrt{d'} \qquad (54)$$

### I.1   TIME COMPLEXITY

**GAT** As noted by Veličković et al. (2018), the time complexity of a single GAT head may be expressed as $\mathcal{O}\left(|\mathcal{V}|dd' + |\mathcal{E}|d'\right)$. Because of GAT's static attention, this computation can be further optimized, by merging the linear layer $\boldsymbol{a}_1$ with $\boldsymbol{W}$, merging $\boldsymbol{a}_2$ with $\boldsymbol{W}$, and only then compute $\boldsymbol{a}_{\{1,2\}}^\top \boldsymbol{W}\boldsymbol{h}_i$ for every $i \in \mathcal{V}$.

**GATv2** require the same computational cost as GAT's declared complexity: $\mathcal{O}\left(|\mathcal{V}|dd' + |\mathcal{E}|d'\right)$: we denote $\boldsymbol{W} = [\boldsymbol{W}_1 \| \boldsymbol{W}_2]$, where $\boldsymbol{W}_1 \in \mathbb{R}^{d' \times d}$ and $\boldsymbol{W}_2^{d' \times d}$ contain the left half and right half of the columns of $\boldsymbol{W}$, respectively. We can first compute $\boldsymbol{W}_1\boldsymbol{h}_i$ and $\boldsymbol{W}_2\boldsymbol{h}_j$ for every $i, j \in \mathcal{V}$. This takes $\mathcal{O}\left(|\mathcal{V}|dd'\right)$.

Then, for every edge $(j, i)$, we compute $\text{LeakyReLU}\left(\boldsymbol{W} \cdot [\boldsymbol{h}_i \| \boldsymbol{h}_j]\right)$ using the precomputed $\boldsymbol{W}_1\boldsymbol{h}_i$ and $\boldsymbol{W}_2\boldsymbol{h}_j$, since $\boldsymbol{W} \cdot [\boldsymbol{h}_i \| \boldsymbol{h}_j] = \boldsymbol{W}_1\boldsymbol{h}_i + \boldsymbol{W}_2\boldsymbol{h}_j$. This takes $\mathcal{O}\left(|\mathcal{E}|d'\right)$.

Finally, computing the results of the linear layer $\boldsymbol{a}$ takes additional $\mathcal{O}\left(|\mathcal{E}|d'\right)$ time, and overall $\mathcal{O}\left(|\mathcal{V}|dd' + |\mathcal{E}|d'\right)$.

**DPGAT** also takes the same time. We can first compute $\boldsymbol{h}_i^\top \boldsymbol{Q}$ and $\boldsymbol{h}_j^\top \boldsymbol{K}$ for every $i, j \in \mathcal{V}$. This takes $\mathcal{O}\left(|\mathcal{V}|dd'\right)$. Computing the dot-product $\left(\boldsymbol{h}_i^\top \boldsymbol{Q}\right) \left(\boldsymbol{h}_j^\top \boldsymbol{K}\right)^\top$ for every edge $(j, i)$ takes additional $\mathcal{O}\left(|\mathcal{E}|d'\right)$ time, and overall $\mathcal{O}\left(|\mathcal{V}|dd' + |\mathcal{E}|d'\right)$.

### I.2   PARAMETRIC COMPLEXITY

|                    | GAT          | GATv2         | DPGAT          |
|--------------------|--------------|---------------|----------------|
| Official           | $2d' + dd'$  | $d' + 2dd'$   | $2dd_k + dd'$  |
| In our experiments | $2d' + dd'$  | $d' + dd'$    | $2dd'$         |

Table 19: Number of parameters for each GNN type, in a single layer and a single attention head.

All parametric costs are summarized in Table 19. All following calculations refer to a single layer having a single attention head, omitting bias vectors.

**GAT** has learned vector and a matrix: $\boldsymbol{a} \in \mathbb{R}^{2d'}$ and $\boldsymbol{W} \in \mathbb{R}^{d' \times d}$, thus overall $2d' + dd'$ learned parameters.

**GATv2** has a matrix that is twice larger: $\boldsymbol{W} \in \mathbb{R}^{d' \times 2d}$, because it is applied on the concatenation $[\boldsymbol{h}_i \| \boldsymbol{h}_j]$. Thus, the overall number of learned parameters is $d' + 2dd'$. However in our experiments, to rule out the increased number of parameters over GAT as the source of empirical difference, we constrained $\boldsymbol{W} = [\boldsymbol{W}' \| \boldsymbol{W}']$, and thus the number of parameters were $d' + dd'$.

**DPGAT** has $\boldsymbol{Q}$ and $\boldsymbol{K}$ matrices of sizes $dd_k$ each, and additional $dd'$ parameters in the value matrix $\boldsymbol{V}$, thus $2dd_k + dd'$ parameters overall. However in our experiments, we constrained $\boldsymbol{Q} = \boldsymbol{K}$ and set $d_k = d'$, and thus the number of parameters is only $2dd'$.

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
