# OpenReview forum: "How Attentive are Graph Attention Networks? "
_ICLR.cc/2022/Conference — ICLR 2022 Poster_

### Official Review · Reviewer_QN6d · 2021-10-29

**Correctness:** 4
**Technical Novelty And Significance:** 4
**Empirical Novelty And Significance:** 3
**Recommendation:** 8
**Confidence:** 4

**Details Of Ethics Concerns:**

There are no ethical concerns in this paper.

**Main Review:**

The novelty of the proposed model is very simple; the authors just switch the order of two operations (multiplication with attention weights and the activation). However, I recommend accepting this paper since the implications from its theoretical and empirical results can be significant to the graph neural network community.

Many recent papers are designing variants of attention and applying them for various domains, but no paper clearly answers why the attention looks like what they proposed and why it works. We cannot say this paper is totally free from this criticism, but it can be the first step to finding the optimal attention architecture. I believe that future research of designing attentions can leverage the principle and practices in this work.

*Questions:* First, the authors state that DPGAT (in Transformer) is strictly weaker than GATv2. Then, can we get a better version of the Transformer if we replace its attention with GATv2’s? Here, I am not demanding the full experiments on GATv2-Transformer on NLP tasks, just asking what GATv2 authors think about this. Second, what properties of graphs characterize the simplicity of the task (or the necessity of different rankings of each node)? That is, what properties of datasets cause the different performance gap between GATv2 and GAT by datasets (besides an average degree in link prediction)? For example, the proposed DictionaryLookup task seems to be simple but requires using different rankings. GATv2 outperforms GAT on ogbn-arxiv with a small margin (<0.3%p), but on ogbn-proteins with a relatively large margin (nearly 6.0%p on one head). Third, the classical citation benchmarks are considered to be easy-to-overfit, but we know GAT does not outperform baselines on PubMed. How does GATv2 perform on the PubMed dataset with the public split?

*Suggestions:* It would be nice if the authors report the statistical significance test results on the tables.


**Summary Of The Paper:**

This work proposes GATv2, the improved variant of the graph attention network (GAT). To demonstrate that GATv2 is more expressive than the original GAT, the authors (1) give theoretical justifications with the notion of static/dynamic attention, (2) conduct experiments on the synthetic dataset which GAT cannot learn, but GATv2 can, (3) demonstrate that GATv2 outperforms GAT on various real-world node-, link-, and graph-level prediction benchmarks.


**Summary Of The Review:**

The proposed GATv2 model is simple but more expressive than the original GAT. The authors’ claim is justified both theoretically and empirically. I would recommend the acceptance since the principles and practices presented by this paper can be useful for future research to design attentional models.

---

> ### Author Response · Authors · 2021-11-11
> **Response to Reviewer QN6d**
>
>
> Thank you for your detailed review and for your kind words!
>
> We were pleased to read that "**the implications from its theoretical and empirical results can be significant to the graph neural network community**" and that you "**believe that future research of designing attentions can leverage the principle and practices in this work**".
>
> Please see our answers below. We would love to address additional questions during the discussion period, if anything is unclear.
>
> > The authors state that DPGAT (in Transformer) is strictly weaker than GATv2. Then, can we get a better version of the Transformer if we replace its attention with GATv2’s? Here, I am not demanding the full experiments on GATv2-Transformer on NLP tasks, just asking what GATv2 authors think about this
>
> Theoretically, DPGAT is strictly weaker than GATv2, because there are examples for vectors where the definition of dynamic attention does not hold (Appendix E.2). We believe that in practice, Transformers converge to word vectors and hidden states that avoid these pitfall examples, and thus do not suffer from this weakness in practice.
>
> We also hypothesize that dot-product attention has become the common practice in NLP (Luong et al., 2015, Vaswani et al., 2018) because dot-product inductively biases the attention function toward vector similarity. In NLP, attention was originally introduced for sequence-to-sequence problems and designed to capture some notion of similarity (or alignment) between words from the input sequence and words from the output sequence. Since dot-product is more biased to capture similarity, it might be more useful than GATv2's attention in NLP.
>
> However, it is an interesting direction to combine both types of attention mechanisms in Transformers as different attention heads and allow a Transformer to leverage both kinds of attention. We think that it may be beneficial, as you suggest.
>
>
> > What properties of graphs characterize the simplicity of the task (or the necessity of different rankings of each node)? That is, what properties of datasets cause the different performance gap between GATv2 and GAT by datasets (besides an average degree in link prediction)? For example, the proposed DictionaryLookup task seems to be simple but requires using different rankings. GATv2 outperforms GAT on ogbn-arxiv with a small margin (<0.3%p), but on ogbn-proteins with a relatively large margin (nearly 6.0%p on one head)
>
> Ideally, we would have liked to tell which attention mechanism to use, in advance.
> At first, we hypothesized that the gap between GATv2 and GAT correlates with the homophily/heterophily of the graph, but we did not find such a clear correlation.
> We also hypothesized that the gap correlates with the dataset's average node degree, or the graph's clustering coefficient, but these also were not always true.
> We conclude that the "simplicity" of the task, or the necessity of different rankings for each node, does not depend _only_ on the characteristics of the graph, but also on the characteristics of the needed information in every node, which is more difficult to measure.
>
> We designed the DictionaryLookup problem to serve as the simplest "extreme case" for measuring the dynamicity of an attention mechanism, but we agree that there is still room for future work to design better measurements for the required dynamicity of a graph task.
>
> This is a great question and an interesting direction that we plan to explore in future work.
>
> > The classical citation benchmarks are considered to be easy-to-overfit, but we know that GAT does not outperform baselines on PubMed. How does GATv2 perform on the PubMed dataset with the public split?
>
> We evaluated GATv2 and GAT on PubMed with the public split.
> Note that the exact code of Veličković et al. (2018) for PubMed is not available, so we reproduced the settings described in Section 3.3 of Veličković et al. (2018).
> We performed a grid search of hyperparameters around the settings of Veličković et al. (2018) of 648 overall configurations for each GNN type, and trained each configuration 100 times as Veličković et al. (2018):
>
> ```
> grid = {
>         'dropout': [0.4, 0.6, 0.8],
>         'bias': [True, False],
>         'share_weights': [True, False],
>         'use_residual': [True, False],
>         'num_layers': [0, 1, 2],
>         'heads': [1, 4, 8],
>         'out_feature': [8, 16, 32],
>         'early_stopping': [100],
>         'l2_lambda': [0.001],
> }
> ```
>
>
> Overall, we could not reproduce the exact same numbers as Veličković et al. (2018), but we gave equally fair chances to both GAT and GATv2:
>
> | Model    | Acc           |
> | -----------| ------------- |
> | GAT       | 78.1 ± 0.59  |
> | GATv2   | **78.5** ± 0.38  |
> | p-value  | <0.0001       |
>
>
> It is important to note that PubMed has only **60 training nodes**, which hinders expressive models such as GATv2 from exploiting their approximation and generalization advantages.

---

> > ### Author Response · Authors · 2021-11-11
> > **Response (2/2)**
> >
> > > Suggestions: It would be nice if the authors report the statistical significance test results on the tables.
> >
> > Thank you for this suggestion! we will include p-values in the next version.

---

> > > ### Comment · Reviewer_QN6d · 2021-11-25
> > > **Thank you for your response**
> > >
> > > Dear authors.
> > >
> > > I just read your response, and I am satisfied with it. I still think this is a good paper. I have not read other reviewers' comments, but I will read them before the reviewer discussion. We will contact you if we have any questions at this stage.
> > >
> > > Plus, it would be great if you include what test do you run for this statistical significance in the paper.

---

> > > > ### Author Response · Authors · 2021-11-25
> > > > **Response to reviewer QN6d**
> > > >
> > > > Thank you for reading our response!
> > > >
> > > > We were pleased to read that you are “**satisfied with our response and think this is a good paper**”.
> > > >
> > > > We used an unpaired one-sided t-test. We are aware of the fact that this may not be ideal in this case, but it is a simple and commonly used test. If you think that another test would be a better fit, we would love to add it to the results.

---

### Official Review · Reviewer_jXKx · 2021-11-02

**Correctness:** 3
**Technical Novelty And Significance:** 3
**Empirical Novelty And Significance:** 3
**Recommendation:** 6
**Confidence:** 3

**Main Review:**

Strengths:

1. a new type of GAT model, that can have very borader impact.

2. The design is super simple as shown in Eq.(6) and Eq(7).

3. They also show some mathmatical analysis, and the results is good.

Weakness:

1. The theorem is not clear for me, J_max leads to maximal values of its attentions distribution. Why says that the \alpha compute only static attention?

2. Any example for he continues function \withhat{g} in Eq.(9)?

**Summary Of The Paper:**

In this paper, the author mainly redesign the Graph Attention Network, and they show that the new model can capture the dynamic attention instead of static attention in the original GAT.

**Summary Of The Review:**

In summary, I think the algorithm is this paper is simple yet effective, just swap some of operators in the orignal graph attention network. The impacts are broad. The differences of the dynamic attention and static attention can be further discussed.

---

> ### Author Response · Authors · 2021-11-11
> **Response to Reviewer jXKx**
>
> Thank you for your detailed review and for your kind words!
>
> We were pleased to read that our paper presents  "***a new type of GAT model, that can have a very broad impact***". We agree, and we think that our paper is of crucial importance to any researchers and practitioners who use or build on graph attention networks.
>
> Please see our response below. We would love to address additional questions during the discussion period, if anything is unclear.
>
> > The theorem is not clear for me, $j_{max}$ leads to maximal values of its attention distribution. Why does it say that the $\alpha$ computes only static attention?
>
> In the proof of Theorem 1,
> the claim that $\alpha$ computes only static attention is derived directly from the definition of *static attention* (Definition 3.1):
>
> ...
> there exists a "highest scoring" key $j_{f} \in \left[n\right]$
> 	such that for every query $i \in \left[m\right]$ and key $j \in \left[n\right]$ it holds that
> 	$f\left(q_i, k_{j_{f}}\right) \geq f\left(q_i, k_{j}\right)$.
> ...
>
> The $j_{max}$ of the proof is the $j_f$ that is required in the definition. $j_{max}$ satisfies that for every query node $i$ and key $j$: $\alpha_{i{j_{max}}} \geq \alpha_{ij}$, and thus $\alpha$ computes only static attention.
>
> Please let us know if anything is still unclear.
>
> > What is the continuous function $\widetilde{g}$ in Eq.(9)? (Appendix A)
>
> The function $\widetilde{g}$ is a function that we define for ease of proof, because the universal approximation theorem is defined for continuous functions, and we only need the scoring function of GATv2 $e$ to approximate the mapping $\varphi$ in a finite set of points. So, we need the attention function $e$ to approximate $g$ (from Equation 8) in some specific points. But, since $g$ is not continuous, we define $\widetilde{g}$ and use the universal approximation theorem for $\widetilde{g}$. Since $e$ approximates $\widetilde{g}$, $e$ also approximates $g$ in our specific points, as a special case.
>
> Specifically, we only require that $\widetilde{g}$ will be identical to $g$ in specific $n^2$ points $\\\{ \left[h_i \|\| h_j\right] \mid i,j\in \left[n\right] \\\}$. For the rest of the input space, we don't have any requirement on the value of $\widetilde{g}$, except for maintaining the continuity of $\widetilde{g}$.
>
> There exist infinitely many such possible $\widetilde{g}$ for every given set of keys, queries and a mapping $\varphi$, and there exist many multivariate interpolation methods of $\\\{ \left[h_i \|\| h_j\right] \mid i,j\in \left[n\right] \\\}$  to construct such a specific function (although the concrete functions are not needed for our proof).
>
> **Please, let us know if anything is unclear or if you have any further questions**.

---

### Official Review · Reviewer_BePW · 2021-11-02

**Correctness:** 3
**Technical Novelty And Significance:** 2
**Empirical Novelty And Significance:** 3
**Recommendation:** 5
**Confidence:** 5

**Main Review:**

The experiments show some improvement, but the theory has flaws:

The theory assumes that there is a fixed set of keys shared by all queries,
but this assumption cannot fit GAT.
In GAT, the softmax normalization is applied to each node's adjacent neighbors,
which means each node will only attend to its adjacent neighbors.

The example in Figure 1 and Section 4.1 is a **complete** bipartite graph.
In this special case, queries have the same set of keys, and the theory works,
but the special example is far from the real-world datasets.
How do you define static and dynamic attention when different nodes have different key sets?

**Summary Of The Paper:**

This paper analyzes the limitation of GAT by pointing out that GAT computes a limited kind of attention: static attention.
This paper then introduces a simple fix by modifying the order of operations and proposes GATv2: a dynamic attention variant.
Experiments show that GATv2 can outperform GAT.

**Summary Of The Review:**

The empirical improvement cannot be properly established on static and dynamic attention

---

> ### Author Response · Authors · 2021-11-11
> **Response to Reviewer BePW**
>
> Thank you for taking the time to review our paper!
>
> We think that all your questions are addressable within this discussion period. Please see our response below. We would love to address additional questions during the discussion period, if anything is unclear.
>
>
> > The theory assumes that there is a fixed set of keys shared by all queries, but this assumption cannot fit GAT. In GAT, the softmax normalization is applied to each node's adjacent neighbors, which means each node will only attend to its adjacent neighbors.
>
> > How do you define static and dynamic attention when different nodes have different
> key sets?
>
> First, please note that the properties of “dynamic” and “static” attention are properties of an **architecture**. The theory uses a graph with a joint set of keys as a **distinguishing criterion** for separating dynamic and static attention. This criterion classifies the GAT architecture as having “static attention” and GATv2 as having "dynamic attention".
>
> In real-world datasets, the theory has a direct impact for any subgraph with keys that share more than one query, and each query needs to attend to the keys differently. Such subgraphs are very common in a variety of real-world domains. While such cases occur in any domain, they are easier to imagine in examples such as products purchased together (ogbn-products) programming-language related tasks (e.g., VarMisuse), etc.
>
> **Please, let us know if anything is unclear or if you have any further questions**.
>
> We believe that our theory does not have any "flaws" as you suggest.

---

> > ### Comment · Reviewer_BePW · 2021-11-23
> > **The response is still not convincing**
> >
> > I appreciate the authors' timely response and update, but the response cannot convince me.
> >
> > > First, please note that the properties of “dynamic” and “static” attention are properties of an architecture.
> >
> > This statement is not accurate. Instead, properties of dynamic attention or static attention **depend on data**, because Definition 3.1 and Definition 3.2 assume that all queries have the same set of keys. For GAT, as I have already pointed in my initial review, the softmax normalization is applied to each node's adjacent neighbors, so **intrinsically GAT allows different nodes to attend to different key sets**. **Only when GAT is applied to a complete graph will GAT exhibit static attention**. In other words, **the theory only points out a special failure case of GAT and is not able to judge the expressive power of GAT in general cases**.
> >
> > > In real-world datasets, the theory has a direct impact for any subgraph with keys that share more than one query.....Such subgraphs are very common in a variety of real-world domains....
> >
> > This might be an explanation, but it needs a more formal setup and investigation. I don't think it is trivial to generalize to subgraphs.
> >
> > I insist on my rating.

---

> > > ### Author Response · Authors · 2021-11-24
> > > **Response to reviewer BePW**
> > >
> > > Thank you for your quick response and for engaging in a discussion!
> > > We are happy to see that you have read our paper thoroughly, and we appreciate you investing time in thinking about it.
> > >
> > > The main point of the paper is recognizing the severe expressivity problem that exists in GAT and proposing a simple fix. Although the GAT architecture has become extremely popular (5600+ citations!) across various domains, the crucial limitation of GAT that we present in this paper is not known to the vast majority of the researchers and practitioners who use it.
> > >
> > > > the theory only points out a special failure case of GAT and is not able to judge the expressive power of GAT in general cases
> > >
> > > Difference in expressivity between architectures is determined by considering *difficult cases*. For example, the GIN paper (Xu et al., ICLR'2019) showed that GIN is as powerful as the Weisfeiler-Lehman (WL) test, and in particular, *more powerful than GCN*. Nonetheless, in a simple "chain" graph, there might be no difference between GCN and GIN. Does it mean that the difference in expressivity between GIN and GCN depends on the data? No, there is an *inherent* expressiveness difference between GIN and GCN. This inherent expressiveness difference is realized in the difficult cases (e.g., Figure 3 of Xu et al.).
> > >
> > > As another example, Garg, Jegelka & Jaakkola (ICML'2020) show additional cases of graphs that standard message-passing GNNs cannot express properly, and cases that even GNNs with port-numbering (CPNGNN) and geometric information (DimeNet) fail with. These cases are very specific, but the limitations of the GNNs are architectural and general.
> > >
> > > Similarly, in simple, degenerate cases, (for example - a "chain" graph) there might be no difference between GAT and GATv2. The difference between GAT and GATv2 is expressed in the difficult cases, in which there are mappings ($\varphi$) that GAT cannot learn, and GATv2 can.
> > >
> > > To formalize this limitation, we defined the notion of "static" and "dynamic" attention, which allows us to demonstrate how GAT is limited, and how this limitation is solved in GATv2. We believe that these definitions are correct and do not have "flaws". We definitely agree that finer-grained separations of attention mechanisms and graph cases are valuable directions for future work. However, since this is the first work to explicitly highlight this major limitation, we decided to focus on the most significant difference.
> > >
> > > > Are "dynamic" and "static attention" properties of the architecture or the data?
> > >
> > > "Dynamic attention" and "static attention" are defined in Definitions 3.1 and 3.2 for a given set of keys and queries, but our theorems 1+2 hold "...*for any set of node representation*...". That is, GAT computes *static* attention for *all* possible sets of nodes, and GATv2 computes *dynamic* attention for all possible sets of nodes.

---

> > > > ### Comment · Reviewer_BePW · 2021-11-29
> > > > **The response is against the paper itself**
> > > >
> > > > **Please note that the response is against the paper itself**
> > > >
> > > > > Difference in expressivity between architectures is determined by considering difficult cases...
> > > >
> > > > Unfortunately, this paper considers only one special case: a complete bipartite graph
> > > >
> > > > A complete bipartite graph is certainly not difficult nor general enough

---

> > > > > ### Author Response · Authors · 2021-11-29
> > > > > **Response to reviewer BePW**
> > > > >
> > > > > Thank you again for engaging in a discussion! We think that your suggestions here bring up an idea that can really simplify the presentation of the DictionaryLookup problem. Please see our response below.
> > > > >
> > > > > > A complete bipartite graph is not difficult
> > > > >
> > > > > Exactly! The novel insight in our paper is that even such non-difficult cases - **are difficult for GAT!**
> > > > >
> > > > > > A complete bipartite graph is not general
> > > > >
> > > > > The results in the paper **apply to any graph**. They do not require a complete nor bipartite graph at all, and the criterion itself also applies in other even more general cases.
> > > > > A complete bipartite graph is a simple example through which the impact is easily evaluated and demonstrated.
> > > > >
> > > > > To make it absolutely clear, every DictionaryLookup example graph (Figure 2) can be equivalently presented as multiple simpler graphs, where every graph has only a *single* "query node" in its upper row, with the same "key nodes" in its bottom row.
> > > > >
> > > > > For example, the graph in Figure 2 would be **equivalently** presented as 4 new graphs, where one graph has only `(A,?)`, and the other three graphs will have the  `(B,?)`,`(C,?)`,`(D,?)` query nodes. This shows how every (*simplified*-)DictionaryLookup example is very general: it is just a single query node that is connected to multiple neighbors:
> > > > >
> > > > > ```
> > > > > (A,?)
> > > > > ├── (A,4)
> > > > > ├── (B,3)
> > > > > ├── (C,2)
> > > > > └── (D,1)
> > > > >
> > > > > (B,?)
> > > > > ├── (A,4)
> > > > > ├── (B,3)
> > > > > ├── (C,2)
> > > > > └── (D,1)
> > > > >
> > > > > (C,?)
> > > > > ├── (A,4)
> > > > > ├── (B,3)
> > > > > ├── (C,2)
> > > > > └── (D,1)
> > > > >
> > > > > ...
> > > > > ```
> > > > >
> > > > > This *simplified*-DictionaryLookup is **completely equivalent** to the case presented in Figure 2. Actually, this was our initial implementation, which empirically resulted in the exact same results. We thought that presenting this as a bipartite graph would form a more direct connection to the theory and make it easier for the readers. Unfortunately, it seems to have confused readers rather than help them. We therefore agree to present the problem in this **equivalent** form if you think that would be easier for the readers to understand and generalize.

---

### Official Review · Reviewer_MFSQ · 2021-11-08

**Correctness:** 3
**Technical Novelty And Significance:** 2
**Empirical Novelty And Significance:** 2
**Recommendation:** 5
**Confidence:** 4

**Main Review:**

The explanations and justifications of the proposed methods can be improved. Why is it a problem that, for any query node, the attention function is monotonic with respect to the neighbor (key) scores? As currently presented, the dynamic attention defined in Definition 3.2 seems a convenient choice. Why is it necessary that every key can be selected using a query? Also, if a key is important, why does any query want to ignore this key or decay the key’s score. I am really doubting the method, but just feel the explanations and justifications can be clearer.

The simple fix captured by Equation 7 was claimed to be able to compute dynamic attention as defined in the paper. Is there a deeper reason for changing the order of local operations? Why LeakyReLU first is better?

Some results show that GATv2 seems to outperform GAT on a set of benchmark datasets, but are mixed in other datasets. In node prediction task in Section 4.4, GATv2 sometimes does better with 1 attention head, and other times better with 8 attention heads. Given it is able to compute dynamic attention, shouldn’t more attention heads be better? In link prediction in Section 4.6, no-attention-based methods outperform attention-based methods. Why is attention not needed in these datasets? Also, why do you think some datasets require dynamic attention? Gaining deeper insights to these questions will be really helpful.




**Summary Of The Paper:**

The authors describe a limitation of the GAT, claiming that the ranking of the attention scores is unconditional on the query node. The paper proposes a method named dynamic graph attention by modifying the order of operations.


**Summary Of The Review:**

The proposed method has potential, but the methods needs better explanations and justifications; and the results seems limited and mixed.

---

> ### Author Response · Authors · 2021-11-11
> **Response to Reviewer MFSQ**
>
> Thank you for taking the time to review our paper!
>
> We think that all your questions are addressable within this discussion period. Please see our response below. We would love to address additional questions during the discussion period, if anything is unclear.
>
> > Why do we need every key to be able to be selected using an appropriate query?
>
> Great question! In many domains, different keys may have different importance to different queries. This need is very frequent in many real-life domains such as modeling computer programs, combinatorial optimization algorithms, complex structures such as molecules, and other semantically rich domains such as natural language, where the concept of attention was originally introduced (Bahdanau et al., ICLR 2015). When this situation exists in the data, we must have an attention mechanism that is powerful enough to express it.
>
> Our DictionaryLookup problem is a minimal example that demonstrates this need, where every "query node" needs to find and attend to a **different** "key node".
>
> > Why is it a problem that "for any query node, the attention function is monotonic with respect to the neighbor (key) scores"?
>
> If the attention function is monotonic with respect to the key scores,  the query node has no influence on the *choice* of its most attended key. It means that for any query node and a set of keys, the **order** (=ranking) of the key scores does not depend on the query nodes. This severely hurts and limits the expressivity of an attention mechanism, making the attention mechanism almost degenerated.
>
> This also contradicts the original motivation for attention in neural networks, which was designed to be able to compute this exact kind of query-dependent choice of keys (e.g., Bahdanau et al. 2015, Xu et al. 2015 ("Show, Attend and Tell"), Vaswani et al. 2018, Veličković et al., 2018).
>
> > If a key is important, why would a query want to ignore this key or decay its score?
>
> Great question, this is the core point: a key may be important for a specific query, but not important (or less important) for another query, as in our DictionaryLookup problem. In such a case, the other query needs to decay the key's score, to allow another key to have a higher score.
>
> A dynamic attention mechanism such as GATv2 can express such dynamic data, and also express "static" relations where some keys are "globally important" or "globally influential".
>
> > Why LeakyReLU first is better? Is there a deeper reason for changing the order of local operations?
>
> Yes!
>
> In GAT (Equation 6), the two linear layers ($a$ and $W$) are applied **right after each other**, and can thus be collapsed into a **single** linear layer. This makes GAT's attention function "almost linear" (because the nonlinear LeakyReLU is applied "too late", on a scalar, and LeakyReLU is a monotonic function).
>
> In contrast, applying the nonlinearity **between** the linear layers $a$ and $W$  (Equation 7) prevents the linear layers from collapsing, makes GATv2's attention mechanism a **multilayer perceptron** (with a single hidden layer), and thus a **universal approximator**, which can approximate any desired attention pattern.
>
> Note that these are only intuitive explanations; for formal explanations, see the proofs for Theorem 1 in Section 3.2 and Theorem 2 in Appendix A.
>
> > In node prediction task, shouldn’t more attention heads be better for GATv2?
>
> Sometimes GATv2 performs better with a single attention head than with 8 heads. We believe that the reason is that GATv2 is so expressive compared to the "difficulty" of the data, that a single head is sufficiently expressive, and 8 heads overfit.
>
> However, our main goal is to show that the GATv2 architecture itself is more powerful than the standard GAT. Using single/multi-head attention is orthogonal to the choice of attention mechanism.
> The *multi-head* experiments show that GATv2 with a **single** head is usually even better than GAT with **multiple** heads.

---

> > ### Author Response · Authors · 2021-11-11
> > **Response (2/2)**
> >
> > > Why do some datasets require dynamic attention?
> >
> > > Why is attention not needed in the link prediction datasets?
> >
> > As discussed in Section 4.7, some "simpler" datasets have an underlying static ranking of "globally important" nodes. For example, in a social network like Instagram, a celebrity may be always influential for all its followers (who are the queries). In more complex domains such as computer programs and molecules, a node might *not* be influential to another node, even though they are connected in the graph, but *is* influential to another, third node.
> >
> > Intuitively, we believe that the more difficult the task and the more complex the interactions between nodes are - the more benefit a GNN can take from theoretically stronger graph attention mechanisms such as GATv2.
> > In general in machine- and deep learning, a theoretically weaker model might perform similarly or better in practice. One reason can be overfitting - a theoretically stronger model may overfit the training data if the task is "too simple" and does not require such expressiveness.
> >
> > Another option is that attention might not be needed in the link-prediction datasets due to the low node degrees. See also the discussion in Sections 4.6 and 4.7.
> >
> > > GATv2 outperforms GAT on a set of benchmark datasets, but the results are mixed in other datasets
> >
> > Our main goal is to show that GATv2 is more powerful, theoretically and empirically, than the standard GAT.
> > In this sense, the results are **not** mixed:
> > * In all our 11 benchmarks, GATv2 **always** outperforms GAT.
> > * In some benchmarks, attention is not needed, and non-attentive GNNs (such as GCN) outperform both GAT and GATv2.
> >
> > See also a discussion in Section 4.7 and Section 4.6.
> >
> > **Please, let us know if anything is unclear or if you have any further questions**.

---

### Author Response · Authors · 2021-11-17
**Any questions before the end of the discussion period?**

We thank all reviewers for the detailed and insightful comments about our work!

We were pleased to read that “**the implications from its theoretical and empirical results can be significant to the graph neural network community**“, that “**future research of designing attentions can leverage the principle and practices in this work**” (Reviewer QN6d), and that our paper presents “**a new type of GAT model, that can have a very broad impact**” (Reviewer jXKx).

Please let us know if there are additional questions or concerns that we can address before the end of the discussion period.

We would be happy to do any follow-up discussion or address any additional comments.

---

### Author Response · Authors · 2021-11-22
**General comment #2 (2021-11-22)**

We'd like to thank the reviewers again for their valuable feedback. We appreciate the feedback you've given and we agree that the clarifications better support the theory we propose, and so make for a stronger paper.

In this revised version (2021-11-22) we have:
* Emphasized the importance of reordering the local operations of GAT to create GATv2 (Section 3.3, as asked by Reviewer MFSQ).
* Clarified that the DictionaryLookup problem was *designed* as a "minimal working example" that exposes a severe limitation of GAT, and demonstrates our theory in many subgraphs that exist in real-world datasets (Section 4.1, as asked by Reviewer BePW).
* Clarified how Theorem 1 is derived directly from Definition 3.1 (Section 3.2, as asked by Reviewer jXKx).
* Clarified the role of $\widetilde{g}$ in the proof of Theorem 2 (Appendix A, as asked by Reviewer jXKx).
* Included the PubMed dataset as the 12th benchmark in which GATv2 outperforms GAT (Appendix D.3, as requested by Reviewer QN6d), despite the very few available training nodes (60 nodes).
* Clarified that the single/multi-head results emphasize that even a single head of GATv2 outperforms GAT with 8 heads (Section 4.4, as asked by Reviewer MFSQ).
* Included statistical significance test results (Appendix F, as suggested by Reviewer QN6d).

We feel that all the questions and concerns raised by the reviewers are now addressed and answered.
If the reviewers agree, we hope that they would consider raising their score.
If there are still any questions or concerns, we would love to further discuss.

During this discussion period, GATv2 was adopted into PyTorch Geometric, the Deep Graph Library (DGL), and Google's new TensorFlow GNN library. This allows future work to easily reuse and build on top of the ideas and theory presented in our work. The rest of our experimental code will be published when possible, and our experiments are fully reproducible.

Again we thank you for your responses and we look forward to the coming discussion.

---

### Decision · Program_Chairs · 2022-01-20

**Decision:**

Accept (Poster)

**Comment:**

This paper argues that the widely adopted graph attention networks (GAT) have a shortcoming that with the static nature of the attention mechanism, they may fail to represent certain graphs. This paper presents an alternative, GATv2, a simple variant with the same time complexity as GAT but with more expressivity, able to represent the graphs that GAT fails to. This is shown both empirically and theoretically, with various tasks on synthetic as well as standard benchmark graphs.

GATs are of high interest to the ICLR community, and this paper makes fundamental progress in how attention works in GNNs. This is one of the few papers that present both empirical and theoretical analyses, and these findings will motivate others in the community to make further advances in this field.